# Ethylene-Cytokinin Interaction Determines Early Defense Response of Wheat against *Stagonospora nodorum* Berk.

**DOI:** 10.3390/biom11020174

**Published:** 2021-01-28

**Authors:** Svetlana V. Veselova, Tatyana V. Nuzhnaya, Guzel F. Burkhanova, Sergey D. Rumyantsev, Elza K. Khusnutdinova, Igor V. Maksimov

**Affiliations:** 1Institute of Biochemistry and Genetics, Ufa Federal Research Centre, Russian Academy of Sciences, Prospekt Oktyabrya, 71, 450054 Ufa, Russia; tanyawww89@mail.ru (T.V.N.); guzel_mur@mail.ru (G.F.B.); rumyantsev-serg@mail.ru (S.D.R.); elzakh@mail.ru (E.K.K.); igor.mak2011@yandex.ru (I.V.M.); 2Ufa Institute of Biology, Ufa Federal Research Centre, Russian Academy of Sciences, Prospekt Oktyabrya, 69, 450054 Ufa, Russia

**Keywords:** plant–microbe interaction, *Stagonospora nodorum*, necrotrophic effectors, SnTox3, ethylene, cytokinins, crosstalk, reactive oxygen species, NADPH oxidases

## Abstract

Ethylene, salicylic acid (SA), and jasmonic acid are the key phytohormones involved in plant immunity, and other plant hormones have been demonstrated to interact with them. The classic phytohormone cytokinins are important participants of plant defense signaling. Crosstalk between ethylene and cytokinins has not been sufficiently studied as an aspect of plant immunity and is addressed in the present research. We compared expression of the genes responsible for hormonal metabolism and signaling in wheat cultivars differing in resistance to *Stagonospora nodorum* in response to their infection with fungal isolates, whose virulence depends on the presence of the necrotrophic effector SnTox3. Furthermore, we studied the action of the exogenous cytokinins, ethephon (2-chloroethylphosphonic acid, ethylene-releasing agent) and 1-methylcyclopropene (1-MCP, inhibitor of ethylene action) on infected plants. Wheat susceptibility was shown to develop due to suppression of reactive oxygen species production and decreased content of active cytokinins brought about by SnTox3-mediated activation of the ethylene signaling pathway. SnTox3 decreased cytokinin content most quickly by its activated glucosylation in an ethylene-dependent manner and, furthermore, by oxidative degradation and inhibition of biosynthesis in ethylene-dependent and ethylene-independent manners. Exogenous zeatin application enhanced wheat resistance against *S. nodorum* through inhibition of the ethylene signaling pathway and upregulation of SA-dependent genes. Thus, ethylene inhibited triggering of SA-dependent resistance mechanism, at least in part, by suppression of the cytokinin signaling pathway.

## 1. Introduction

In natural growing conditions, plants are permanently in contact with pathogens and respond to their attacks by activating defense mechanisms regulated by a number of interacting signals [1,2]. The current model of the plant immune system proposes that host defenses are opposed to counterattacks by pathogens at several levels [3]. The plant initially identifies pathogen-associated molecular patterns (PAMPs) by pattern recognition receptors (PRRs), leading to the development of basal immunity, known as pattern triggered immunity (PTI) [3]. To overcome PTI and the subsequent defense signaling, pathogens have developed molecules known as effectors, which induce an effector-triggered susceptibility (ETS) [3]. Effectors are directly or indirectly recognized by products of effector-specific resistance genes, which often encode intercellular proteins called nucleotide binding domain and leucine-rich repeat domain proteins. This recognition results in development of effector-triggered immunity (ETI) in biotrophic pathogens [3]. Until recently, very little was known about how plants struggle with necrotrophic pathogens, which, in contrast to biotrophs, obtain nutrients from dead tissue [4]. However, recent studies have revealed that some necrotrophic fungal pathogens produced effector proteins also known as host-selective toxins (HSTs) or necrotrophic effectors (NEs) that interact either directly or indirectly with dominant sensitivity/susceptibility gene products to induce disease [5]. Thus, NEs suppress PTI and use the host’s ETI pathway to develop sensitivity, resulting in NE-triggered susceptibility (NETS) [5,6].

Numerous fungal NEs encoded by SnTox genes are the most important virulence factors of pathogenic fungus *Stagonospora nodorum* Berk. (syn. *Septoria, Parastagonospora;* teleo, *Phaeosphaeria*), the causative agent of Septoria nodorum blotch (SNB) in wheat [7,8]. The interaction in the wheat–*S. nodorum* pathosystem is of the gene-for-gene type [8]. The products of the virulence genes of pathogens (=necrotrophic effectors or host-specific toxins) (SnTox) interact with the products of the susceptibility genes of the host plants (*Snn*) followed by disease development [7]. To date, three effector genes have been identified in the genome of *S. nodorum* (*SnToxA*, *SnTox1*, *SnTox3*) [7]. The effectors, SnToxA, SnTox1, and SnTox3, are quite widespread among strains and isolates and are considered to be the main ones in the pathogen *S. nodorum* [9]. The effectors, SnToxA, SnTox1, and SnTox3, cause necrosis and chlorosis in susceptible wheat genotypes and affect redox metabolism of the host plant [8]. In addition, the role of NEs SnToxA, SnTox1, and SnTox3 in the suppression of PTI and the development of NETS is expected [8].

The development of PTI induces multiple cellular responses, including the generation of reactive oxygen species (ROS) and calcium-dependent or mitogen-activated protein kinase cascade activation, which subsequently leads to the reprogramming of the transcriptome and proteome [10]. Phytohormones play a key regulatory role in these primary immune responses upon activation of PTI. Salicylic acid (SA), jasmonic acid (JA), and ethylene are considered to be the classical plant hormones of immunity [1]. These phytohormones constitute the central regulatory network of plant immunity and interact with growth-related plant hormones such as cytokinins (CKs), auxins, abscisic acid (ABA), brassinosteroids (BRs), and gibberellins (GAs) [1,11].

Ethylene is a plant hormone that not only regulates the processes of plant growth and development, but it is also one of the main phytohormones of plant immunity. However, the role of ethylenein plant defense is ambiguous. It has been shown that activation of the ethylene signaling pathway can lead to both resistance and susceptibility of plants to pathogens [12,13]. It has been suggested that ethylene acts as a positive or negative inducer of resistance depending on the type of pathogen and is a regulator of the interaction between SA- and JA-dependent defense reactions [14]. Crosstalk among different hormonal networks is often observed in plant immunity. The antagonistic nature of the crosstalk between the SA and JA pathways was the first to be shown in plant immune responses [15]. Then, synergism between JA and ethylene, and antagonism between SA and ethylene pathways were also revealed in various pathosystems [11,16]. However, to date, the role of ethylene itself, as well as its relationship with other phytohormones, especially with CKs, in plant–microbe interactions, remains largely uncharacterized.

Recently, the important role of the classical phytohormone CK has been revealed in the development of plant resistance through the regulation of SA-dependent defense reactions, induction of the gene expression of protective proteins, phytoalexins synthesis, and lignification processes [17,18,19,20]. Thus, CK–ABA antagonism has been identify in tobacco as a novel regulatory mechanism to modulate resistance against *Pseudomonas syringae* [19]. Unfortunately, the interaction of ethylene and CK has been studied only in the regulation of growth and development processes [21,22] and in response to abiotic stress factors [21,23], where an antagonistic relationship has been shown between CK and ethylene. Thus, questions regarding the nature of the interaction between ethylene and CK remain open when it comes to plant immunity.

The mechanisms underlying hormonal crosstalk are not fully understood. However, there is a widespread opinion that signaling proteins, such as Nonexpressor of PR Genes1 (NPR1), DELLA, Arabidopsis Response Regulator (ARR), Ethylene Insensitive3 (EIN3) and Ethylene Insensitive3-Like1 (EIL1), transcription factors of WRKY family (transcription factors of WRKY family that contain WRKY domains at the N-terminus, having a conserved heptapeptide sequence WRKYGQK, and a zinc-finger-like motif at the C-terminus) and others, can be considered as hormone crosstalk hubs [11]. Frequently, the same signaling proteins involved in hormonal crosstalk are the targets of the effectors [24]. Effectors secreted by pathogens and providing successful plant colonization by suppressing PTI interfere with the pathways of phytohormones [24]. Manipulation of hormonal pathways by effectors occurs at different levels. Effectors can affect phytohormone biosynthesis or important components of hormonal signaling pathways [24]. SnTox3 has been shown relatively recently to activate biosynthesis of ethylene in plants [25] and further more SnToxA and SnTox3 directly interacted with the PR-1 protein and enhanced the infection of wheat by *S. nodorum* [26].

We have previously shown that ethylene provided favorable conditions for the penetration and growth of *S. nodorum* in the tissues of wheat plants at the initial stage of infection due to the regulation of redox metabolism and a decrease in the generation of H_2_O_2_ [27], and development of the wheat plants resistance against *S. nodorum* was accompanied by an increase in the zeatin content in the leaves [28]. Here, we address the early signaling events in wheat responding to the *S. nodorum* and analyze the role of NE SnTox3 in the suppression of the PTI reaction such as ROS production. To achieve this, we employed an integrated approach using real-time qPCR, ELISA and microscopy methods. We demonstrated that NE SnTox3 induced an ethylene signaling pathway and inhibited biosynthesis, modulated metabolism, and activated oxidative degradation of CKs to regulate ROS production in wheat at the early stages of *S. nodorum* infection. Our results suggest that antagonism of ethylene and CK is directed to the regulation of SA-dependent defense reactions responsible for the development of PTI and early wheat resistance against *S. nodorum*. In addition, we provide novel evidence regarding the role of the CK as a crucial regulator of plant immunity. These findings are useful for understanding the mechanisms underlying the manipulation of plant hormonal signaling pathways by pathogen effectors during the suppression of PTI and the development of ETS (NETS).

## 2. Materials and Methods

### 2.1. Plant and Fungi Materials and Growth Conditions

The objects of the study were two cultivars of common wheat (*Triticum aestivum* L.) contrasting in resistance to *S. nodorum* Berk.: Omskaya 35 (Om35) (resistant) and Kazakhstanskaya 10 (Kaz10) (susceptible), with different allelic states of the susceptibility locus *Snn3-B1* [29]. Wheat seeds were obtained from the Bashkir scientific research Institute of Agriculture of Russian Agricultural Academy. We used two isolates of the fungus *S. nodorum*: Sn4VD (avirulent) and SnB (virulent) (from the collection of Institute of Biochemistry and Genetics, Ufa Federal Research Centre, Russian Academy of Sciences, Ufa, Russia). All *S. nodorum* isolates were maintained on potato-glucose agar (PGA) at 21 °C for a 12 h photoperiod. Plants were hydroponically grown on 10% solution of Hoagland–Arnon nutrient medium in a KS-200 SPU growth chamber (Russia) at 20/24 °C (night/day) at an irradiance of 146 W/m^2^ FAR (Osram lamps L 36 W/77, Osram Licht AG, Munich, Germany) and with a 16 h photoperiod for six days.

### 2.2. Experimental Design

All experiments were carried out on intact six-day-old seedlings, with the exception of experiments evaluating the resistance of genotypes, which were performed on the separated first leaves by the lawns method [27]. In some cases, parts of 6-days-old seedlings placed in separate vessels were sprayed with 2 mM solution of 1-methylcyclopropene (1-MCP) to study the role of ethylene. The compound was prepared from its SmartFresh precursor (AgroFresh Inc., Philadelphia, PA, USA). The procedure was performed no later than 5 min after preparing the solution when gaseous 1-MCP arose [30]. Other plants were sprayed with 1.5 mM solution of 2-chloroethylphosphonic acid (ethephone, ET) (Merck KGaA, Sigma-Aldrich, Darmstadt, Germany) in separate vessels [31]. To study the role of cytokinins, parts of 6-day-old seedlings were sprayed with 2.5 μM solution of *trans*-zeatin (Merck KGaA, Sigma-Aldrich, Darmstadt, Germany). The immune-stimulating concentration of *trans*-zeatin was selected according to [32]. All solutions contained the wetting agent Tween-20 (0.02%). Control plants were sprayed with a solution containing only the wetting agent Tween-20 (0.02%). The volumes of all solutions allowed full moistening of leaves. In all cases, vessels were closed with caps and transferred into the darkness. During the experiments by the lawns method, the first leaves of the 6-day-old seedlings were separated and placed in Petri dishes on wet cotton wool containing 0.004% benzamidazole (10–12 leaves/dish) [27]. Then, leaves were sprayed with 1-MCP, ET, or *trans*-zeatin, and the Petri dishes were covered and placed in the darkness. All treatments were performed 24 h prior to inoculation with *S. nodorum* isolates. In the case of studying the role of salicylic acid (SA), a presowing treatment was performed by soaking the seeds for 3 h in a 0.05 mM SA solution (Merck KGaA, Sigma-Aldrich, Darmstadt, Germany). Then, the plants were grown as described above. The biologically effective concentration of SA was selected according to [33,34,35].

To study the effect of the necrotrophic effector SnTox3, two *S. nodorum* isolates were used. Previously, it was shown that the virulent isolate SnB expressed SnTox3 and caused severe damage in susceptible genotypes [29]. Avirulent isolate Sn4VD did not express SnTox3 and did not cause significant damage in both resistant and susceptible genotypes [29]. After the treatments described above, seedlings were either sprayed with a spore suspension of *S. nodorum* isolates with the addition of a wetting agent Tween-20 (0.02%), or the drops of the fungal spore suspension were applied to the separated leaves in the amount of 5 μL per leaf. Concentration of suspension was 1 × 10^6^ spores mL^−1^ in both cases. Then, the vessels with seedlings and Petri dishes with the separated leaves were closed with caps and transferred to the controlled conditions of the growth chamber. The development of SNB symptoms was observed on separated wheat leaves placed in Petri dishes for 8 days. The lesion areas were registered using an SP-800UZ Image Stabilization camera (Olympus, Bekasi, Indonesia) on the eighth day. The damage zones were measured with the ImageJ 1.44 computer program (rsbweb.nih.gov/ij/download.html) and expressed as a percentage of the total leaf area.

To study biochemical characteristics, the shoots of intact wheat seedlings that underwent various treatments were fixed in liquid nitrogen 1, 6, 12, 24, 48 and 72 h after inoculation with *S. nodorum* virulent isolate SnB. To study gene expressions, the shoots of intact wheat seedlings were fixed in liquid nitrogen 6 and 24 h after inoculation with *S. nodorum* isolates SnB and Sn4VD. In the case of studying the transcriptional activity of the *TaWRKY13* and *TaRR21* genes, plants were fixed 15 min, 3, and 6 h after inoculation with *S. nodorum* virulent isolate SnB. Cytokinins were quantified in shoots 24 h after inoculation with *S. nodorum* isolates SnB and Sn4VD using an enzyme-linked immunosorbent assay (ELISA) after their solvent partitioning and purification. The variants of treatments and the number of repetitions are indicated in the tables and figures.

### 2.3. Isolation of RNA and Performing the Quantitative Real-Time Polymerase Chain Reaction (qPCR)

Total wheat RNA was extracted using TRIzol™ Reagent (Sigma, Germany) according to the manufacturer’s instructions. The potential contaminating DNA was digested with DNaseI (Synthol, Moscow, Russia). First-strand cDNA was synthesized using the M−MLV reverse transcriptase (Fermentas). Oligo(dT)15 was used as a primer, and the reverse transcription reagents were incubated at 37 °C for 1 h in a total volume of 25 μL. After ten-fold dilution, 2 μL of the synthesized cDNA was used for quantitative real-time polymerase chain reaction (qPCR). The primers for qPCR were designed based on the cDNA sequence (Appendix A, see Appendix A). Quantitative PCR was performed by polymerase chain reaction in real time using a set of predefined reagents, EvaGreenI (Synthol, Moscow, Russia), and a CFX Connect real-time PCR Detection System device (BioRad Laboratories, Hercules, CA, USA). The qPCR program was as follows: 95 °C for 5 min; 40 cycles of 95 °C for 15 s, 60 °C for 20 s, and 72 °C 30 s. After the final PCR cycle, a melting curve analysis was conducted to determine the specificity of the reaction (at 95 °C for 15 s, 60 °C for 1 min, and 95 °C for 15 s). The efficiency of each primer pair was determined using 10-fold cDNA dilution series in order to reliably determine the fold changes. To standardize the data, wheat gene *TaRLI* (RNaseL inhibitor-like) (GenBank Accession No. AY059462) was used as an internal reference for the real-time qPCR analysis. The quantification of gene expression was performed using a CFX Connect real-time PCR Detection System (BioRad Laboratories, USA). All reactions, including the nontemplate control, were performed three times. The threshold values (CT) generated from the CFX Connect real-time PCR Detection System software tool (Applied Biosystems, Foster City, CA, USA) were employed to quantify the relative gene expression using the comparative threshold (delta CT) method. Three independent biological replicates were performed for each experiment.

### 2.4. Biochemical Parameters

To measure the hydrogen peroxide (H_2_O_2_) production and the activity of redox enzymes (peroxidase (POD) and catalase (CAT)), plant material (1:5 weight/volume) was fixed in liquid nitrogen and then it was homogenized in 0.05 M solution of Na-phosphate buffer (PB), pH 6.2 and incubated at 4 °C for 30 min. Supernatants were separated by centrifugation at 15,000× *g* for 15 min (5415 K Eppendorf, Hamburg, Germany). Concentration of H_2_O_2_ in the supernatant was determined using xylenol orange in the presence of Fe^2+^ at 560 nm by the method of [36]. POD activity was determined by a micromethod in 96-well plates (Corning-Costar, Glendale, AZ, USA) by the oxidation of (o-) phenylenediamine in the presence of H_2_O_2_ at 490 nm on a Benchmark Microplate Reader spectrophotometer (Bio-Rad Laboratories, Hercules, CA, USA) [27]. The enzyme activity was expressed in optical density/mg protein per minute. CAT activity was determined by a micromethod based on the ability of H_2_O_2_ to form a stable colored complex with molybdate salts [27]. Optical density was measured at 405 nm on a Benchmark Microplate Reader spectrophotometer. CAT activity was calculated using a calibration curve and expressed in μM H_2_O_2_/(mg protein per min). Protein content was determined by the Bradford method.

### 2.5. Visualization of H_2_O_2_, Superoxide Radical, and Fungal Mycelium in Wheat Tissues

The local generation of superoxide radicals and H_2_O_2_ in infected leaf tissues was determined by vital staining with solutions of nitroblue tetrazolium (NBT) (0.1%) (Sigma, Germany) and diaminobenzidine (DAB) (1 mg mL^−1^) (Sigma, Germany), respectively, prepared as described in [12,37,38,39]. For this purpose, 6 and 24 h after inoculation with *S. nodorum*, leaf sections were incubated in vacuum for 60 min at 20 °C, and then the leaves were fixed in 96% ethanol and transferred into a 50% glycerol solution. To visualize the mycelium of *S. nodorum*, 24 h after inoculation with the pathogen leaf sections were fixed in 96% ethanol and then stained with 1.0% aniline blue solution in 1% lactic acid. As a result, the structures of the fungus became blue-violet [39]. To register the accumulation of superoxide radical and H_2_O_2_ and the development of the pathogen mycelium, a BZ8100E digital microscope (Keyence, Osaka, Japan) was used.

### 2.6. Determination of Cytokinines

Shoots of ten plants (approximately 1 g) per one biological replication were homogenized and cytokinins were extracted with 80% ethanol (1:10, weight/volume) for 16 h at 4 °C. The extract was separated by 20-min centrifugation at 4000× *g* in an Avanti J-E centrifuge (Bekman Coulter, Bray, CA, USA) and evaporated to obtain aqueous residue. Different forms of cytokinins present in an aliquot of aqueous residue were concentrated on a C18 column (Waters Corporation, Milford, MA, USA), eluted with 5 mL of 80% ethanol and then evaporated to dryness. Cytokinin bases and their derivatives from the dry residue were separated by thin layer chromatography on silufol plates (Merck KGaA, Fluka, Darmstadt, Germany) in the system of solvents butanol: ammonium hydrate: water (6:1:2) according to the work of [40]. This procedure enabled separation and assay of cytokinin nucleotide (Rf 0–0.1), cytokinin glucosides (Rf 0.1–0.2), riboside of zeatin (ZR, Rf 0.4–0.5), isopentenyladenosine (iPA, Rf 0.5–0.6), zeatin (Z, Rf 0.6–0.7), and isopentenyladenine (iP, Rf 0.7–0.8). The material from different zones was eluted with 0.1 M PB, pH 7.4 for 16 h. Then, silica gel was removed by 10-min centrifugation at 10,000× *g* in an Eppendorf 5415 K centrifuge. In the supernatant, phytohormone was assayed by means of ELISA as earlier described using specific antibodies [40]. Anti-trans-ZR and anti-iPA sera were used for the assay of cytokinins of Z and iP types, correspondingly [40]. Their specificity has been described previously [40,41]. Since O-glucosides have very low affinity to the antibodies used in this work, treatment with β-glucosidase (0.02 mg enzyme from Sigma, USA per milliliter sample from 0.5 g of fresh leaves) was carried out for 4 h at 37 °C, pH 5.0, to release immunoreactive zeatin for zeatin glucoside quantification as described [42]. The reliability of the hormone immunoassay has been confirmed using a dilution test and through comparison with the data obtained with the results of high performance liquid chromatography (HPLC) in combination with mass spectrometry [40,42].

### 2.7. Statistics

Experiments were performed three times with three replicates for each treatment, except for the measurements of infected areas, which were performed in no less than 30 biological replications for each experiment. One replicate contained shoots of ten plants in the case of ELISA, and it contained shoots of five plants in the case of qPCR and biochemical assay of enzyme activity and H_2_O_2_ production. Leaf segments of ten plants were fixed for each treatment for histochemical localization of H_2_O_2_ and superoxide radicals. Microphotographs represent results of a typical variant from a series of experiments. Experimental data were expressed as means ± SE, which were calculated in all treatments using MS Excel. Analysis of variance (ANOVA) was used to calculate the least significance difference (LSD) at *p* < 0.05 to discriminate means.

## 3. Results

### 3.1. Biosynthesis and Signaling Pathway of Ethylene in Wheat Is Activated by the Effector Stagonospora nodorum Berk

The Om35 and Kaz10 cultivars were selected as the SnTox3-insensitive and SnTox3-sensitive cultivars, respectively. Plants were inoculated with two *S. nodorum* isolates, one was virulent, expressing SnTox3 (SnB) and the other was avirulent and did not express SnTox3 (Sn4VD) [29]. An incompatible interaction or resistance reaction was detected when the SnTox3-insensitive cultivar, Om35, was inoculated with a virulent isolate SnB or when cultivars Om35 and Kaz10 were inoculated with an avirulent isolate, Sn4VD. A compatible interaction or susceptibility response was developed when the SnTox3-sensitive cultivar Kaz10 was inoculated with a virulent isolate SnB. Compatible interaction (Kaz10/SnB) was characterized by the development of extensive lesion zones, covering up to 70% of the total leaf area, with typical spots of chlorosis and necrosis with numerous pycnidia, which are organs of asexual reproduction (Figure 1). In all incompatible interactions (Om35/SnB, Kaz10/Sn4VD, Om35/Sn4VD), the coverage of damage zones was from 0.5 to 12% of the total leaf area and these were characterized by the absence of mycelium, pycnidia, and chlorosis and consisted of necrotic zones, which may indicate a cessation of pathogen growth (Figure 1).

Pretreatment of plants with ET increased the virulence of the isolate SnB in both Kaz10 and Om35 cultivars. The reaction of cultivar Om35 to SnB was similar to the reaction of cultivar Kaz10 with this isolate; the damage of leaf zones was up to 70% of the total area, but there was less chlorosis and pycnidia than in the Kaz10 response to SnB infection (Figure 1). In the Kaz10/SnB/ET combination, large lesion areas with active sporulation of the pathogen were detected, covering up to 90% of the total leaf area (Figure 1).

Pretreatment of plants with ET did not increase the virulence of the isolate Sn4VD. Pretreatment of plants with 1-MCP reduced the virulence of the SnB isolate on both Kaz10 and Om35 cultivars, but to a greater extent on the Kaz10 variety (Figure 1). In the Kaz10/SnB/1-MCP and Om35/SnB/1-MCP combinations, minimal lesions with small necrosis were found. The proportion of area damaged per leaf was about 2%. Pretreatment of plants with 1-MCP did not affect the virulence of the isolate Sn4VD (Figure 1). These results show the negative role of ethylene in the development of wheat resistance against *S. nodorum* due to the sensitivity of the host plant to the NE SnTox3 and the connection of SnTox3 with the ethylene signaling pathway.

To check the effect of NE SnTox3 on the biosynthesis and signaling pathway of ethylene in plants, both cultivars Kaz10 and Om35 were pretreated with either ET or 1-MCP and were inoculated with the isolate SnB. We studied the expression of genes involved in ethylene biosynthesis (aminocyclopropane (ACC) synthase (*TaACS1*) and ACC oxidase (*TaACO*)), as well as genes of the ethylene signaling pathway (*TaEIL1* and *TaPIE1*) (Figure 2). The gene of transcription factor (TF) *TaEIL1* (Ethylene-Insensitive3-Like1—EIN3-Like1) is an orthologue of the Arabidopsis gene *AtEIN3* encoding the main regulatory factor of ethylene signaling. The gene of TF Pathogen-Induced ERF1 (*TaPIE1*) is an orthologue of the Arabidopsis gene *AtERF1*, which is involved in the primary response to ethylene, regulating the expression of second-order regulatory genes (Appendix A, see Appendix A).

Analysis of the transcriptional activity of the *TaACS*, *TaACO*, *TaEIL1*, and *TaPIE1* genes showed an increase in the mRNA abundance of these genes in the SnTox3-sensitive variety Kaz10 and in plants treated with ET (Kaz10/SnB/ET, Om35/SnB/ET) (Figure 2). In SnTox3-sensitive plants, Kaz10 the transcript levels of ethylene biosynthesis genes *TaACS, TaACO*, and the *TaPIE1* gene increased approximately 3–4 times, and the mRNA abundance of the *TaEIL1* gene increased 14–18 times at 24 hpi (Figure 2).

Interestingly, the accumulation of mRNA of genes responsible for ethylene biosynthesis and signaling pathway in SnTox3-insensitive plants in the Om35/SnB/ET combination was lower than that in SnTox3-sensitive plants; the transcript levels of all genes increased only from 1.5 to 3 times at 24 hpi (Figure 2). These results show that the induction of the ethylene signaling pathway was strongly dependent on gene-for-gene interactions. In plants treated with 1-MCP, both SnTox3-sensitive (Kaz10) and SnTox3-insensitive (Om35), suppression of transcript accumulation of ethylene biosynthesis and signaling pathway genes was found (Figure 2).

### 3.2. SnTox3 Suppresses Oxidative Burst through the Ethylene Signaling Pathway by Regulating the Work of NADPH Oxidase and Provides the Growth of the Pathogen

Components of the host plant redox metabolism were studied at the initial stage of infection with *S. nodorum* in order to determine the role of NE SnTox3 and the contribution of the ethylene signaling pathway to the suppression of oxidative burst.

In our work, two peaks of H_2_O_2_ generation were found in SnTox3-insensitive plants Om35 infected with the isolate SnB at 6 and 24 hpi (Figure 3B). In the SnTox3-sensitive Kaz10 cultivar, no significant increase in the content of H_2_O_2_ was detected at the initial stage of infection (Figure 3A). Ethephon application suppressed the accumulation of H_2_O_2_ in both SnTox3-sensitive and SnTox3-insensitive plants at 6 and 24 hpi (Figure 3A,B). However, in the SnTox3-sensitive Kaz10 plants, the decrease in H_2_O_2_ content was more pronounced (Figure 3A). Conversely, exogenous 1-MCP application induced accumulation of H_2_O_2_ in leaves at 6 and 24 hpi in both Kaz10 and Om35 cultivars (Figure 3A,B). These results suggest that the change in the H_2_O_2_ content in infected wheat plants in our experiments completely depended on the activation or inhibition of the ethylene signaling pathway induced by SnTox3.

We have studied the transcription of *TaRbohD* and *TaRbohF* genes required for ROS accumulation in the plant defense response against pathogen infection [43,44]. The transcript levels of the *TaRbohD* gene at 6 hpi and of the *TaRbohF* gene at 24 hpi increased up to 4- and 5-fold in SnTox3-insensitive Om35 plants and plants treated with 1-MCP and infected with SnB (Figure 3C,D). This coincided with the peaks of H_2_O_2_ generation (Figure 3B). Moreover, the transcript level of the *TaRbohD* gene decreased at 24 hpi in these plants, although it remained higher than in the control (Figure 3C). The opposite nature of the transcription of *TaRbohD* and *TaRbohF* genes was found in SnTox3-sensitive Kaz10- and ET-treated plants (Figure 3C,D). The slight increase in mRNA content of the *TaRbohD* gene was observed after 6 h of infection, followed by a significant increase up to 4- and 5-fold at 24 hpi (Figure 3C). The transcript level of the *TaRbohF* gene did not change at 6 hpi and significantly decreased at 24 hpi compared to the control (Figure 3D). Importantly, treatment with 1-MCP of SnTox3-sensitive plants only partially restored the transcript level of the *TaRbohF* gene compared to SnTox3-insensitive plants infected with SnB (Figure 3D). Treatment with ET of SnTox3-insensitive plants only partially inhibited the transcript level of *TaRbohF* gene compared to SnTox3-sensitive plants infected with SnB (Figure 3D). Thus, our results suggest that SnTox3 downregulated *TaRbohF* and upregulated *TaRbohD* at 24 hpi. SnTox3 regulated *TaRbohF* transcription in ethylene-dependent and ethylene-independent manners.

Inoculation of both Kaz10 and Om35 cultivars with the avirulent isolate Sn4VD led to a slight two-fold increase in the transcript level of *TaRbohD* at 6 hpi, regardless of the treatment with either ET or 1-MCP, while the abundance of *TaRbohD* mRNA decreased at 24 hpi in all variants except those treated with ET (Figure 3F). The transcript level of the *TaRbohF* gene increased up to 1.5-fold at 24 h after Sn4VD inoculation in all interaction variants (Figure 3E). These results suggest that the ethylene signaling pathway is capable of completely regulating *TaRbohD* gene transcription, but only partially regulating transcription of the *TaRbohF* gene.

Then, we analyzed local accumulation of superoxide radicals and H_2_O_2_ at the site of pathogen invasion, as well as the proliferation of fungal mycelium with histochemical staining methods to gain clearer insight into the cellular defense reactions of wheat against *S. nodorum* (Figure 4). Spore germination and penetration of fungal hyphae through the stomata occurred within 6 hpi (Figure 4A). Staining with NBT revealed a very strong accumulation of superoxide radical in hyphae-containing epidermis and mesophyll cells under stomata 6 h after infection with the isolate SnB of SnTox3-insensitive Om35 plants and plants treated with 1-MCP (Figure 4A). In this case, the mycelium is poorly developed on the plants leaves (Figure 4C).

Staining with DAB of SnTox3-insensitive Om35 plants and plants treated with 1-MCP revealed a very strong accumulation of H_2_O_2_ in epidermis and mesophyll cells at the sites of pathogen penetration at 24 hpi (Figure 4B). In this case, H_2_O_2_ performed a direct biocidal function. On the contrary, a small accumulation of superoxide radical in hyphae-containing mesophyll cells under stomata was observed in SnTox3-sensitive Kaz10 plants and ET-treated plants at 6 hpi (Figure 4A). This resulted in the rapid propagation of the pathogen mycelium on the leaf surface and the appearance of numerous sites of penetration into the plant at 24 hpi (Figure 4C). Intensive mycelium growth was accompanied by almost complete absence of H_2_O_2_ accumulation in stomata, epidermis and mesophyll cells in the places of fungus penetration at 24 hpi (Figure 4B). These observations suggest that restriction of fungal proliferation occurred mainly at the stage of infection spread in the mesophyll due to the intense local accumulation of the superoxide radicals and H_2_O_2_. Conversely, the ethylene signaling pathway induced by SnTox3 created favorable conditions for the penetration and propagation of *S. nodorum* in wheat leaf tissues by suppressing the oxidative burst at the initial stage of infection.

### 3.3. Cytokinins Enhance the Oxidative Burst, Limit the Growth of the Pathogen and Trigger the Salicylate Signaling Pathway

An important function of CKs, both in the processes of growth and development, and under the influence of stress factors, is the regulation of plant redox metabolism [45]. One of the most important effects of CKs is a delay in senescence, and they also have an antioxidant effect on plants [46]. Interestingly, at high concentrations, CKs induce growth inhibition, control programmed cell death (PCD) in the development and senescence program [45], and also increase the production of ROS during plant responses to stress factors of various natures [38,47].

Before inoculation with the virulent isolate of SnB, the SnTox3-insensitive Om35 and SnTox3-sensitive Kaz10 were treated with either *trans*-zeatin (2.5 μM) or salicylic acid (SA) (50 μM) to study the role of CKs and SA in the development of wheat defense responses against *S. nodorum*. Treatment with *trans*-zeatin or SA reduced the virulence of SnB in both Om35 and Kaz10 cultivars. Leaves of both cultivars treated with *trans*-zeatin developed small lesion zones, which consisted only of necrosis and occupied about 9% of the total leaf area (Figure 5A,B). The response of both cultivars to SA treatment was similar to the response upon *trans*-zeatin treatment; however, necrosis spots were smaller in area and additionally small chloroses were observed on the leaves of both cultivars (Figure 5A,B). These data uncover the role of CKss and SA as activators of induced resistance against *S. nodorum*.

Treatment with either *trans*-zeatin or SA induced two peaks of H_2_O_2_ generation 6 and 24 h after infection with the isolate SnB of SnTox3-sensitive Kaz10 cultivar and increased H_2_O_2_ production in the SnTox3-insensitive Om35 cultivar (Figure 5C). The pattern of changes in the transcript levels of *TaRbohD* and *TaRbohF* genes upon inoculation with the SnB of plants of both cultivars treated with either *trans*-zeatin or SA was the same as in SnTox3-insensitive cultivar Om35 (Figure 6A,B). The transcript level of the *TaRbohD* gene was strongly accumulated at 6 hpi, and the mRNA content of the *TaRbohF* gene significantly increased at 24 hpi, which coincided with the peaks of H_2_O_2_ generation (Figure 5C). In addition, the oxidative burst in both cultivars treated with either *trans*-zeatin or SA and infected with SnB was accompanied by a significant increase in the activity of free peroxidases and inhibition of catalase activity, similar to the response of the SnTox3-insensitive Om35 cultivar (Figure 6C,D). In contrast, the compatibility reaction in the SnTox3-sensitive Kaz10 cultivar was characterized by a slight increase in peroxidase activity and a strong increase in catalase activity at an early stage of infection, which could lead to the absence of oxidative burst found in SnTox3-sensitive Kaz10 cultivar (Figure 6C,D). Our results show that cytokinins and SA trigger defense reactions in wheat against *S. nodorum* due to the induction of oxidative burst at an early stage of infection by acting on the enzymes of redox metabolism, such as NADPH oxidase, peroxidase, and catalase.

To understand whether cytokinins induce SA-dependent defense reactions, we examined the transcription of salicylate signaling pathway marker genes *PR-1* and *PR-2* in Om35 and Kaz10 varieties after their treatment with either *trans*-zeatin or SA and infection with virulent isolate SnB. The transcript levels of *PR-1* and *PR-2* genes did not increase in the SnTox3-sensitive cultivar Kaz10 at 24 hpi (Figure 7). The mRNA abundance of *PR-1* and *PR-2* genes increased up to 3-fold in the SnTox3-insensitive cultivar Om35 at 24 hpi (Figure 7). These results clearly indicate that resistance in SnTox3-insensitive cultivar developed in an SA-dependent manner. Treatment with either *trans*-zeatin or SA significantly increased mRNA accumulation of *PR-1* and *PR-2* genes in both SnTox3-sensitive and SnTox3-insensitive cultivars at 24 hpi (Figure 7). Our data suggest that cytokinins induce SA-dependent resistance mechanisms in wheat against *S. nodorum* associated with the development of oxidative burst at the early stages of infection.

### 3.4. SnTox3 and Ethylene Regulate Biosynthesis and Metabolism of Cytokinins During Defense Response of Wheat against S. nodorum

We have previously shown that enhanced resistance of wheat plants against *S. nodorum* was accompanied by an increase in zeatin content in the leaves. Ethephon treatment reduced zeatin content in infected leaves and, conversely, treatment with 1-MCP increased the content of this phytohormone [28]. In this work, we determined the changes in the content of the most biologically active forms of CKs: N6-(Δ2-isopentenyl)-adenine, (iP) N^6^-(Δ^2^-isopentenyl)-adenosine (iPR), zeatin (Z), and zeatin riboside (ZR) in both Kaz10 and Om35 cultivars, treated with either ET or 1-MCP and inoculated with either SnB or Sn4VD. In addition, reversible storage forms of cytokinins (Z-O-glucoside (Z-OG)) and inactive forms (zeatin-9-N-glucoside (Z-9G)) were also measured. In addition, to clarify the mechanism underlying regulation of the active form levels of CKs in wheat infected with *S. nodorum*, we analyzed gene transcription of individual members of the *IPT, CKX, ZOG*, and *GLU* multigene families involved in CK biosynthesis and metabolism. *TaIPT* gene family members encode adenosine phosphate-isopentenyltransferase isoforms that catalyze the synthesis of cytokinins by attaching an isopentenyl radical to the purine ring of adenine; *TaZOG* gene family members encode isoforms of zeatin-O-glucosyl transferases that catalyze the reaction of O-glucosylation (inactivating cytokinins by their conjugation with glucose radical); *TaGLU* gene family members encode β-glucosidase isoforms that catalyze the deglucosylation reaction (the conversion of the bound form of O-glucosides into the active form); *TaCKX* gene family members encode isoforms of cytokinin oxidases that catalyze oxidative degradation of CK [48,49].

In incompatible interactions (Om35/SnB, Om35/Sn4VD, Kaz10/Sn4VD), we observed an increase in the content of all active forms of CK—Z, iP and their ribozides at 24 hpi (Table 1). Amount of CK active forms increased up to 2-fold (Table 1). Along with this, a very high mRNA accumulation of *TaIPT2* and *TaIPT5* genes was observed in incompatible interactions as early as 6 h after infection (Figure 8A,B). In the compatible interaction (Kaz10/SnB), we observed no increase in concentrations of CK active forms at 24 hpi (Table 1). The levels of Z and iPR decreased by 1.5 and 2 times, respectively, compared with uninfected control (Table 1), and a 2-fold decline in the transcript levels of the *TaIPT2* and *TaIPT5* genes was also found in a compatible interaction (Figure 8A,B). These results clearly indicate that the pathogen *S. nodorum* can induce a decrease in CK active forms by inhibiting CK biosynthesis.

Decrease in CK active forms similar to the compatible interaction of Kaz10/SnB was found in ET-treated SnTox3-sensitive Kaz10 plants infected with virulent SnB (Kaz10/SnB/ET) (Table 1). Furthermore, these plants showed a stronger suppression of transcription of *TaIPT2* and *TaIPT5* genes compared to Kaz10/SnB (Figure 8A,B). Importantly, in the case of the compatible interaction of Kaz10/SnB, pretreatment with 1-MCP only partially restored the accumulation of CK active forms (especially Z and iPR) (Table 1) and the transcript levels of the *TaIPT2*, *TaIPT5* genes, compared with the incompatible interaction Om35/SnB (Figure 8A,B). These results suggest that inhibition of CK biosynthesis involves both ethylene-dependent and -independent signaling pathways. Treatment with ET of SnTox3-insensitive Om35 plants infected with virulent SnB (Om35/SnB/ET) prevented accumulation of CK active forms and partially inhibited transcription of the *TaIPT2* and *TaIPT5* genes compared to the incompatible interaction Om35/SnB (Table 1, Figure 8A,B). Importantly, the 1-MCP pretreatment of such plants completely restored the increased levels of CK active forms and transcript levels of *TaIPT2* and *TaIPT5* genes to the values characteristic of incompatible interaction of Om35/SnB (Table 1, Figure 8A,B). This proves only partial involvement of the ethylene signaling pathway in this process. Treatment with ET as well as 1-MCP of SnTox3-sensitive Kaz10 and SnTox3-insensitive Om35 plants infected with avirulent Sn4VD did not affect the accumulation of mRNA of *TaIPT2* and *TaIPT5* genes and insignificantly affected the level of CK active forms (Table 2, Figure 8C,D).

A slight decrease in the amount of CK active forms was found in ET-treated plants infected with Sn4VD, and a slight increase in the amount of CK active forms was found in 1-MCP-treated Sn4VD-infected plants, which was in both cases mainly due to changes in Z concentration (Table 2). Thus, suppression of CK biosynthesis was not found in plants infected with the avirulent Sn4VD nonexpressing SnTox3 gene.

CK metabolism plays an important role in the regulation of the levels of active CKs. Glucosylation is a major step in the metabolism of cytokinins [48]. In incompatible interactions (Om35/SnB, Om35/Sn4VD, Kaz10/Sn4VD), increases in the levels of active CK forms was accompanied by a decrease in Z-OG content, while Z-9G content did not change (Table 1 and Table 2). This correlated with the increase in mRNA abundance of the *TaGLU1-1* and *TaGLU4* genes and a decrease in the transcript level of the *TaZOG2-1* gene (Figure 8). It may suggest a release of CK active forms from bound Z-OG forms and a decrease in the activity of the glucosylation reaction. In a compatible interaction (Kaz10/SnB), s reduction in CK active forms levels was accompanied by significant increase in the Z-9G and Z-OG contents up to 1.5- and 2-fold, respectively (Table 1), which correlated with a very high (up to 12-fold) mRNA accumulation of the *TaZOG2-1* gene and 2-fold downregulation of *TaGLU4* gene compared to uninfected plants (Figure 8A,B). This may suggest a sharp induction of the glucosylation and a decline in deglucosylation.

Ethephone treatment without infection led to about to 2–3-fold accumulation of Z-9G and Z-OG compared to untreated plants (Table 1 and Table 2). This was accompanied by about 3-fold upregulation of the *TaZOG2-1* gene and a slight decline in mRNA abundance of the *TaGLU4* gene (Figure 8A,B). Treatment with 1-MCP prevents the effects of ethylene. 1-MCP-treated noninfected Om35 and Kaz10 plants showed a reduction in Z-OG levels and a decrease in mRNA abundance of *TaZOG2-1* gene (Table 1, Figure 8A,B). These results uncover the role of ethylene in the glucosylation reaction. During infection, treatment with ET also led to 2–3-fold increase in Z-9G and Z-OG (Table 1 and Table 2), a very strong (from 6- to 12–fold) increase in transcript level of *TaZOG2-1*, a decline in mRNA abundance of the *TaGLU1-1* gene, and an extreme (from 2- to 30-fold) decrease in transcripts of the *TaGLU4* gene (Figure 8) in all variants (Om35/SnB/ET, Om35/Sn4VD/ET, Kaz10/SnB/ET, Kaz10/Sn4VD/ET), regardless of the genotype of either pathogen or host. This clearly indicates that ethylene has a role in this process. During infection, treatment with 1-MCP prevented the effect of ethylene. In the 1-MCP-treated SnTox3-sensitive Kaz10 and SnTox3-insensitive Om35 plants infected with either virulent SnB or avirulent Sn4VD, we observed reduced levels of Z-OG, no significant changes in Z-9G concentrations, increase in transcript levels of *TaGLU1-1* and *TaGLU4* genes, and a decline in mRNA abundance of the *TaZOG2-1* gene (Table 2, Figure 8C,D). Taken together, our results reveal that NE of SnTox3 could affect CK metabolism only through activation of the ethylene signaling pathway—i.e., in an ethylene-dependent manner.

The CK level in the plant can be regulated by either decreasing or increasing the oxidative degradation of CK. An extreme increase in the transcript level of the *TaCKX1* gene from 7- to 10-fold has been detected in SnTox3-sensitive Kaz10 plants pretreated with ET or not 24 h after infection with virulent SnB (Figure 8A). Importantly, 1-MCP pretreatment in the compatible Kaz10/SnB interaction only partially removed the effect of ethylene: about 2- to 4-fold mRNA accumulation of the *TaCKX1* gene remained in the Kaz10/SnB/1-MCP variant at 6 and 24 hpi (Figure 8A). In all other variants, in both cultivars upon infection, a similar tendency of the changes in the *TaCKX1* gene transcription was detected—that is, the transcript level increased about two times at 6 hpi and decreased at 24 hpi compared to control plants (Figure 8). These results demonstrate that only the virulent SnB expressing SnTox3 gene was able to activate *TaCKX1* in the susceptible genotype, while treatment with either ET or 1-MCP had a weak effect on *TaCKX1* transcription. Thus, SnTox3-induced *TaCKX1* transcription in both ethylene-dependent and ethylene-independent manner.

### 3.5. Interaction of Signaling Pathways of Ethylene, CK, and SA

We have studied individual elements involved in the signaling pathways of ethylene, CK, and SA under the influence of treatment with ET, 1-MCP, SA, and *trans*-zeatin in plants of the Om35 and Kaz10 cultivars. In this work, we studied the TFs ERF1 (*TaPIE1*) and EIN3 (*TaEIL1*) as elements of the ethylene signaling pathway [50,51], the type-B cytokinin response regulator *TaRR21* which is an ortholog of the *Arabidopsis* gene ARR2 [52], and *TaWRKY13* as a marker of the development of the SA signaling pathway which is an ortholog of the *Arabidopsis* gene *AtWRKY70* [53]. (Appendix A, see Appendix A).

Treatment with *trans*-zeatin and SA downregulated the expression of *TaEIL1* and *TaPIE1* in both SnTox3-sensitive Kaz10 and SnTox3-insensitive Om35 plants infected with virulent SnB, indicating inhibition of the ethylene signaling pathway (Figure 9A,B). A similar nature of the transcription of genes was observed in infected plants of both varieties treated with 1-MCP (Figure 2). In SnTox3-insensitive Om35 plants and plants of both varieties treated with either *trans*-zeatin, SA or 1-MCP, the expression of *TaWRKY13* increased (Figure 9C), suggesting induction of the SA signaling pathway in these plants. In the SnTox3-sensitive Kaz10 cultivar and plants of both cultivars treated with ET, we observed inhibition transcription of the *TaWRKY13* gene (Figure 9C). These results indicate that CK-activated transcription of SA-dependent genes and SnTox3-induced ethylene inhibited this response in plants infected with *S. nodorum*. Analysis of the transcriptional activity of the type-B RR genes showed that cytokinin signaling was not activated in SnTox3-sensitive Kaz10 plants and in plants of both cultivars treated with ET and infected with virulent SnB. The transcript level of the *TaRR21* gene was decreased in these plants (Figure 9D). In SnTox3-insensitive Om35 plants and in plants of both cultivars treated with either *trans*-zeatin or 1-MCP and infected with virulent SnB, the transcription of this gene was activated (Figure 9D). The level of transcripts of the *TaRR21* gene increased about 2–3 times compared to the uninfected control (Figure 9D). These results suggest activation of cytokinin signaling.

## 4. Discussion

Although over the past 15 years significant progress has been made in cloning and characterizing the *S. nodorum* effectors SnToxA, SnTox1, and SnTox3 [8,9] and the dominant wheat susceptibility genes *Tsn1* and *Snn1* [6,54], little is known about the effector responses and hormonal signaling pathways operating in the determination of wheat resistance. The few studies of the transcriptome and proteome of sensitive wheat genotypes have been carried out to investigate the impact of effectors on the plants defense response, showing both the activation and repression of a large number of genes upon toxin infiltration [25,55]. With regard to SnTox3, only one study of this type is known, in which the effect of SnTox3 on primary metabolism, photosynthetic proteins, and genes associated with signaling, redox metabolism, and ethylene synthesis in SnTox3-sensitive wheat genotype was shown using a comprehensive transcriptomic, proteomic, and metabolomic approach [25]. The results of this work revealed many interesting processes in the host plant associated with action to NE SnTox3. However, we now need to functionally characterize some of these processes and define exactly what role they play in disease.

### 4.1. Stagonospora nodorum NE SnTox3 Induces Biosynthesis and Signaling Pathway of Ethylene

In this study, we have used SnTox3-sensitive and SnTox3-insensitive wheat genotypes as well as two *S. nodorum* isolates expressing (SnB) and nonexpressing (Sn4VD) genes encoding the host-specific necrotrophic effector SnTox3 to explore the SnTox3 interference with the hormonal signaling pathways of ethylene, CK, and SA. Our data showed that NE SnTox3 induced *TaACS* and *TaACO* genes involved in ethylene biosynthesis only in sensitive wheat genotypes due to gene-for-gene interactions (Figure 2). Our results are in accordance with the data published previously and obtained using the transcriptomic and proteomic approach [25]. Genes encoding aminocyclopropane synthase (ACS) and ACC oxidase (ACO), both involved in the synthesis of ethylene from S-adenosyl methionine (SAM), were upregulated in SnTox3-infiltrated plants [25]. These data clearly show that SnTox3 induces activation of ethylene biosynthesis in the susceptible wheat genotype. The induction of the ethylene signaling pathway by the SnTox3 effector, manifested in the activation of the main positive regulator of this pathway EIN3 (*TaEIL1*) and the transcription factor ERF1 (*TaPIE1*), which we found in the susceptible wheat genotype is consistent with similar data previously published by other authors [51,56].

Importantly, we found that the biosynthesis and signaling pathway of ethylene were only activated in the susceptible genotype and were not activated in the resistant genotype of wheat. These results suggest that ethylene plays a negative role in wheat defense to *S. nodorum*. This assumption is confirmed by our results obtained using pretreatments with the ethylene-releasing chemical ET or 1-MCP, a well-known ethylene receptor blocker, and is consistent with the data obtained by other authors, in which ethylene caused necrosis in plant tissues and contributed to the following colonization of the host plant by the pathogen [12,51,57].

However, it is well known that treatment of plants with ethylene increases either susceptibility or resistance, depending on the plant–pathogen interaction, and on the conditions of the interaction. Thus, ethylene enhanced the resistance of rice to *Magnaporthe oryzae* [58], but increased the susceptibility of rice to *Cochliobolus miyabeanus* [12] and of wheat to *Pyricularia oryzae* [59]. The role of ethylene has been extensively studied using plant mutants affected in ethylene production or signaling [60]. However, contradictory results were obtained, indicating in some cases that ethylene can function as a virulence factor of certain pathogens and, in other cases, that it acts as a signaling molecule in defense response [60,61]. The ambivalent effect of ethylene can be explained by the following facts. First, ethylene regulates programmed cell death (PCD) both during the PTI and disease development. Second, the effects of ethylene at different stages of infection might also be quite different. Third, ethylene acts in conjunction with other phytohormones in synergistic or antagonistic interactions, on which the effect of ethylene may depend [62].

### 4.2. Ethylene and Cytokinins Regulate ROS Production in Wheat Plants Infected with S. nodorum

Our results showed that NE SnTox3 was involved in the development of necrosis in wheat tissues, appearing 4–6 days after infection, in an ethylene-dependent manner, which was confirmed by the treatment of plants with ethephon (Figure 1). However, we were interested in the roles of SnTox3 and ethylene in the suppression of plant defense reactions at the early stages of infection before the appearance of visible symptoms. Therefore, our study focused on researching the role of ethylene and its interaction with CK and SA in the regulation of ROS production during PTI and at the early stages of infection (24 hpi) of wheat plants with the pathogen *S. nodorum*.

#### 4.2.1. Ethylene Promotes Penetration and Growth of the *S. nodorum* Fungus in Wheat Tissues

ROS production triggered by PAMPs in the apoplast through the activation of NADPH oxidases and peroxidases leads to PTI-dependent induction of programmed cell death (PCD) and inhibition of pathogen growth [10]. The function of H_2_O_2_ in suppressing the growth of pathogens is fully understood [10]. Plant ROS are powerful weapons against pathogens. However, pathogens, using effectors, can avoid unwanted cytotoxic ROS accumulation [10]. We found that a strong local accumulation of H_2_O_2_ and superoxide radicals in the epidermis and mesophyll cells at the sites of pathogen penetration stopped the growth of *S. nodorum* mycelium (Figure 4). The SnTox3-induced ethylene promoted a weak accumulation of superoxide radical and inhibition of H_2_O_2_ production, which led to the proliferation of fungal hyphae in wheat tissues (Figure 4). Similarly, inhibition of mycelium development in pathogen was found in transgenic plants with impaired synthesis or reception of ethylene, [12,16], and the inhibition of ethylene reception by 1-MCP led to an increase in H_2_O_2_ content in the apoplast [63]. It has also been shown that low concentrations of ROS are inducers of morphogenesis in fungi, contributing to their enhanced growth and development [64], and pathogens use phytohormonal signaling pathways in plants to regulate the ROS level [13,65]. Additionally, a direct dependence of the growth rate and spread of the pathogen on the production of ethylene and low ROS concentrations was shown in tobacco plants infected with *Phytophthora parasitica* var nicotianae [57].

Our results show that the rapid production of H_2_O_2_ in resistant SnTox3-insensitive plants and in plants treated with 1-MCP to disrupt ethylene signaling can inhibit the proliferation of fungal mycelium and allow the plant to induce defense mechanisms to effectively stop the pathogen’s penetration into the mesophyll. This suggests that the *S. nodorum*, despite a purely necrotrophic lifestyle, can have a short biotrophic phase in the epidermis, during which it is sensitive to H_2_O_2_-dependent protection, and the function of NE SnTox3 is to create favorable conditions for fungal growth and plant colonization by suppressing oxidative burst at an early stage of infection in an ethylene-dependent manner.

#### 4.2.2. Ethylene Inhibited Biphasic H_2_O_2_ Production and Cytokinins Induced It at Early Stages of Infection with *S. nodorum*

ROS induced by PAMPs also act as signaling molecules, mediating responses of plant cells to pathogen attacks [10,43,66]. The H_2_O_2_ molecule is uncharged, relatively long-living, and can penetrate through membranes; therefore, it is the most suitable candidate for the role of a secondary messenger [43]. We found that the accumulation of H_2_O_2_ with two peaks at 6 and 24 hpi was characteristic of resistant SnTox3-insensitive plants, while the biphasic character of H_2_O_2_ production was absent in susceptible plants and plants treated with ethephon (Figure 3).

Biphasic ROS production or “ROS wave” has been reported in many incompatible plant–pathogen interactions [57,67] as well as in case of abiotic stresses [68]. The first (early) peak occurs within minutes and hours, and the second (long term) one appears within a day after the onset of the stimulus [66]. Modern studies have shown that the first (early) peak in ROS production can trigger a cascade of subsequent reactions, form a long-term ROS signal, and is an important factor in disease resistance [66]. The induction of hormonal signaling pathways is also consistent with the phase character of H_2_O_2_ accumulation [69,70]. It has been previously shown that SA is closely related to the induction and enhancement of the first phase of H_2_O_2_ production and the further development of systemic resistance [69]. Our recent publication showed the induction of SA signaling and increases in expression of the marker genes *PR-1* and *PR-2* during incompatible interactions and inhibition of SA signaling during compatible interactions and under the influence of ethylene in the wheat–*S. nodorum* pathosystem [28]. In this work, pretreatment with SA of SnTox3-sensitive plants induced the biphasic character of H_2_O_2_ production with peaks at 6 and 24 hpi with SnB; an increase in transcripts of the SA pathway marker genes of the *PR-1* and *PR-2* was also found (Figure 7). An assumption was made in which ROS and SA function together in a self-amplifying feedback loop, in which ROS induce SA accumulation and SA subsequently enhances ROS accumulation [71].

Interestingly, the same biphasic H_2_O_2_ production and induction of transcription of the *PR-1* and *PR-2* genes were found in SnTox3-sensitive plants pretreated with *trans*-zeatin (Figure 5 and Figure 7), suggesting activation of the SA pathway by *trans*-zeatin which is consistent with findings of other authors. [17]. Such a positive effect of *trans*-zeatin on the ROS generation in plants is not a surprise because the role of various forms of exogenous CKs (kinetin, 6-benzylaminopurine, *trans*-zeatin) in the induction of ROS generation during the regulation of growth and development processes, as well as responses to stress factors, was shown in various plant and cell cultures [45,47,72]. Thus, our data clearly indicate that cytokinins induce an SA-dependent resistance mechanism in wheat against *S. nodorum* associated with biphasic H_2_O_2_ production at the early stages of infection, while ethylene inhibits this defense mechanism. Finally, the antagonism of CK and ethylene manifested itself at the level of ROS production and led to the opposite influence on the SA-dependent resistance mechanism.

#### 4.2.3. SnTox3, Ethylene and Cytokinins Regulate the Work of Redox Enzymes in Wheat Plants Infected with *S. nodorum*

Apoplast-secreted peroxidases [73] and membrane-bound NADPH oxidases [74] perform significant roles in plant oxidative bursts [43]. We studied two isoforms of wheat NADPH oxidase, RBOHD and RBOHF, which are not comprehensively studied, akin to the *AtRboh* Arabidopsis genes. Nevertheless, there are works showing that the *TaRbohD* and *TaRbohF* genes, which are homologues of the Arabidopsis genes *AtRbohD* and *AtRbohF* and activated under biotic stress, are involved in the production of ROS, and the early peak of H_2_O_2_ production depends on *TaRbohD* [75]. Our results showed that the two isoforms of NADPH oxidase, RBOHD and RBOHF, perform different functions during the defense response of wheat against *S. nodorum*. The incompatible interaction was characterized by an earlier transcript accumulation of the *TaRbohD* gene and a late increase in mRNA abundance of the *TaRbohF* gene, which coincided with the two peaks of H_2_O_2_ production (Figure 3 and Figure 6). Our data are consistent with the results described in the literature for most pathosystems, which indicates a qualitative (spatial or temporal) difference in the ROS produced by each *AtRboh* [57,75,76,77]. The use of various mutants of the model Arabidopsis plants showed that *AtRbohD* is responsible for rapid ROS production during PTI in the first minutes and hours after plant recognition of PAMPs and its activation is PAMP-mediated [44]. The high levels of ROS in the first peak induce *AtRbohF*, which affects intracellular oxidative stress, intercellular signal transmission, and genome reprogramming—i.e., development of systemic resistance [77]. Moreover, SA is required to maintain *AtRbohF* expression and *AtRbohF* ensures sustained SA accumulation [77].

The obtained results show that SA and CK induced early activation of *TaRbohD* transcription and late activation of *TaRbohF* transcription (Figure 6). This caused an incompatibility reaction in susceptible plants and the development of resistance along the SA-dependent pathway, as indicated by the induction of expression of SA pathways marker genes *PR-1* and *PR-2* (Figure 7). On the contrary, SnTox3 inhibited *TaRbohF* transcription specifically through gene-for-gene interactions in ethylene-dependent and ethylene-independent manners, which is confirmed by the treatment of plants with either ethephone or 1-MCP (Figure 3). This led to extensive lesions and the absence of SA-dependent defense reactions, suggesting that *TaRbohF* performs functions similar to *AtRbohF* in induction of the SA signaling pathway and limits lesions [77].

The observed slight increase in the *TaRbohD* gene transcripts at 6 hpi during compatible interactions and under the influence of ET treatment was not associated with oxidative burst and can explain the small accumulation of superoxide radicals in mesophyll cells under stomata, which ensured the penetration and growth of the pathogen. Our data showed that the massive increase in the *TaRbohD* gene transcripts at 24 hpi in SnTox3-sensitive plants was not associated with the H_2_O_2_ production and the development of defense reactions (Figure 3). However, these results can be explained by the reports demonstrating the ability of *AtRbohD*-dependent ROS production to suppress salicylate-mediated cell death at the sites of pathogen penetration, thereby ensuring the spread of infection [78]. Our data strongly suggest an ethylene-dependent way of *TaRbohD* transcription regulation by the SnTox3 effector in wheat at the early stages of *S. nodorum* infection, which is consistent with the literature data on the regulation of *AtRbohD* expression by ethylene [79].

The class III peroxidases are the most ubiquitous plant enzymes. They form an important part of ROS homeostasis in plant–microbial interactions, and occupy one of the key positions in plant defense against pathogens [80]. Our recent publication showed that ET treatment reduced the activity of peroxidases and the transcription of the anionic peroxidase gene in the resistant cultivar Om35 infected with *S. nodorum*, which was accompanied by decreased ROS production [28]. Interestingly, treatment with 1-MCP completely canceled the ethylene effect on the peroxidases activity during infection [28]. Our data coincide with works proving the inhibitory effect of ethylene on peroxidases under the action of a stress factor [81,82,83,84]. The role of CKs in the regulation of redox enzymes under influence of various stress factors, including biotic ones, has not been sufficiently studied; the data found in the literature are scarce, but in some studies it has been shown that CKs increased activity of peroxidases involved in the ROS generation during pathogenesis [47]. Thus, our results on the increase in peroxidase activity under influence of treatment with *trans*-zeatin are consistent with some literature data.

It has been established that some mechanisms of a decrease in the H_2_O_2_ concentration during pathogenesis are associated with catalase activation [66]. Catalases enhance the virulence of fungal pathogens, including *S. nodorum*, by reducing the ROS concentration in the infected zone and suppressing oxidative burst [85]. We have previously shown that increased catalase activity was a characteristic feature of wheat plants treated with ET and infected with *S. nodorum*, being the cause of low H_2_O_2_ production in wheat at the early stages of infection [28]. The role of ethylene in the regulation of catalase activity has not been well understood, especially under biotic stress, but some studies have shown that ethylene is able to activate catalase [81,82,86]. Importantly, SA is known to inhibit catalase activity leading to an increase in the H_2_O_2_ concentration and the development of an oxidative burst, which plays an important role in the system defense antipathogenic reactions of plants [87]. In our work, both SA and *trans*-zeatin treatment led to a strong decrease in catalase activity in wheat at the early stages of *S. nodorum* infection (Figure 6). This was accompanied by biphasic H_2_O_2_ production (Figure 5).

Thus, in the wheat–*S. nodorum* pathosystem, the antagonism of ethylene on the one hand and CK and SA on the other hand at the early stages of infection manifested itself in the effect on redox enzymes—NADPH oxidase, peroxidase, and catalase.

### 4.3. SnTox3 and Ethylene Inhibited Biosynthesis, Modulated Metabolism, and Activated Oxidative Degradation of Cytokinins

One of the important approaches decoding the role of phytohormones in plant immunity is the study of phytohormone crosstalk. In the last decade, the important role of the interactions of CK with SA [17,88] and ABA [19] with auxins [89,90] in plant immunity has been established. Unfortunately, CK–ethylene crosstalk in the defense response has not been studied. Cytokinins are the most important class of phytohormone stimulants that promote active metabolism and plant growth [46,91]. Previously, cytokinins were considered in the plant–pathogen interactions, only as hormones produced by pathogens and necessary for their nutrition and development [92]. The role of CK in the induction of plant defense had been recognized only in the late 2000s [17,93]. Currently, data have been accumulated on the role of CK in the development of plant resistance to biotrophic [94], hemibiotrophic [17,95], and necrotrophic [96,97] pathogens of bacterial or fungal origin, as well as viruses [98]. Nevertheless, the regulatory mechanism and targets of cytokinins during defense responses to pathogens are still elusive. It has been shown that CK can contribute to plant resistance through the regulation of SA-dependent defense reactions, including upregulation of a number of genes encoding protective proteins [17,18,88,93], as well as through the SA-independent pathway leading to the synthesis of phytoalexins and lignification processes [23,95]. In this work, we have found that resistance reactions involved in incompatible interactions are characterized by an increase in the content of active CK brought about by de novo synthesis of CK, their release from bound forms and the absence of an increase in their oxidative degradation. Susceptibility reactions that occur during compatible interactions involve a decline in the content of active CK due to a decrease in the CK synthesis, increase in the irreversible oxidation of CK, and conversion of active CK into inactive forms (Z-9G and Z-OG).

#### 4.3.1. SnTox3 and Ethylene Regulated Cytokinin Biosynthesis in Wheat Plants Infected with *S. nodorum*

First of all, our results clearly indicate that the incompatible interaction was characterized by an increased level of CK active forms in wheat leaves at the initial stage of infection (24 hpi) associated with the induction of oxidative burst and the triggering SA-dependent defense reactions. The compatible interaction was characterized by a directed decrease in the contents of CK active forms due to the regulation of CK biosynthesis, glucosylation, and oxidative degradation by the SnTox3 effector (Table 1). In plants, isopentenyltransferases (IPTs) are considered to be the rate limiting enzyme in cytokinin biosynthesis [48]. We found that the pathogen *S. nodorum* suppressed transcription of *TaIPT* genes in wheat using NE SnTox3 specifically by gene-for-gene interactions in ethylene-dependent and ethylene-independent manners (Figure 8). In the present work, it was shown for the first time that NE SnTox3 interfered, possibly indirectly, in the hormonal pathway of CKs and suppressed their biosynthesis. One of these indirectly mediated mechanisms was the ethylene signaling pathway, which is activated by the SnTox3 effector. It has been known that CKs in some cases play a negative role in the regulation of ethylene biosynthesis. For example, delays in flowering correlating with the increases in CK levels and delays in ethylene biosynthesis were observed in petunia [99]. The possibility that ethylene may affect CK biosynthesis is also being considered, although there is little empirical confirmation for this hypothesis as yet [100]. However, there are reports showing an increased expression of the *AtIPT3* gene and accumulation of CK in roots in ethylene-insensitive *etr1-1* mutants [101], which may indirectly indicate the possible inhibition of CK biosynthesis by ethylene.

#### 4.3.2. SnTox3 and Ethylene Activated Oxidative Degradation of Cytokinins in Wheat Plants Infected with *S. nodorum*

Some studies suggested that pathogens are able to alter CK levels in host plants through the effect on cytokinin oxidases/dehydrogenases (CKXs) [97,102,103] involved in the irreversible oxidation of CKs [104]. Our results showed that pathogen *S. nodorum* induced transcription of the *CKX1* gene in wheat using NE SnTox3 specifically by gene-for-gene interactions in ethylene-dependent and ethylene-independent manners, which correlated with a low level of active CKs in wheat leaves. Our results are consistent with the data obtained on Arabidopsis plants infected with the soilborne fungus *Verticillium longisporum*, which induced transcription of the *AtCKX* genes leading to a decrease in the Z level and the development of necrosis and chlorosis in the leaves [97]. However, it has not been determined as to whether this resulted from the action of effectors secreted by the pathogen. Wan et al. (2019) showed that the effector AvrL567-A from the flax rust fungus *Melampsora lini* interacted with a flax cytosolic cytokinin oxidase, *LuCKX1.1*, increased its catalytic activity, and reduced the cytosolic cytokinin levels, thereby promoting the colonization of the plant [103]. Moreover, indirect activation of CKX is possible through interference with the hormonal pathways of the plant. In the literature, there are data showing activation of CKX by ethylene during organ abscission [105]. Auxins and ABA are also capable of activating CKX during drought [106] or infection [102]. We speculate that the ethylene-independent mechanism of CKX activation could be via the hormonal pathways IAA or ABA.

#### 4.3.3. SnTox3 and Ethylene Modulated Metabolism of Cytokinins in Wheat Plants Infected with *S. nodorum*

The glucosylation reaction (conjugation to sugars) is one of the effective ways to reduce the level of active CKs in plants. Glucosylation of CK at the N3, N7, and N9 positions of the purine moiety forms N-glucosides and their synthesis is practically irreversible. Glucosylation of CK at the hydroxyl group of the side chains of tZ, DZ, and cZ results in production of O-glucosides the process being reversible. O-glucosides are considered as temporarily storage forms of CKs; from which active forms are released by deglycosylation catalyzed by β-glucosidase (GLU) [48,49]. The glucosylation reaction has been well studied during senescence processes [49,107]. A dramatic increase in *TaZOG* expression observed during leaf senescence was accompanied by accumulation of CK O-glucosides in leaves [49]. In one of the few studies, accelerated conversion of active cytokinins into O-glucosides was shown in plants treated with ethylene leading to faster Petunia corolla senescence [107]. In another work, a positive effect of ethylene on the irreversible glucosylation of active CK to 9-N-glucosides was detected [101]. In our experiments, showing upregulation of *TaZOG2-1* gene and inhibition of *TaGLU4* gene transcription by ET led to an increase in the Z-9G and Z-OG contents and a decrease in active CK in infected leaves, which is consistent with the literature data. However, we have shown for the first time that the glucosylation reaction was induced, and deglucosylation reaction was suppressed by NE of SnTox3 only in an ethylene-dependent manner. In the incompatible interaction, on the contrary, the deglucosylation reaction was induced and the CK biosynthesis was activated. We speculate that these processes could have been triggered by PAMPs.

### 4.4. Ethylene Induced by the Effector SnTox3 Suppresses the SA Signaling Pathway and Cytokinins Trigger it in Wheat Plants Infected with S. nodorum

Over the past decade, the molecular mechanisms mediating CK–ethylene crosstalk at various levels of biosynthetic and metabolic pathways, as well as their complex interactions with growth processes and in response to environmental stressors, have been actively studied [22,100,108,109,110,111,112]. Nevertheless, it remains unclear how individual signals are integrated into a unified plant response to a certain stimulus.

Our results show that resistance was associated with activation of the CK signaling pathway. This led to the induction of SA signaling pathway genes and suppression of the transcription of ethylene response genes. The susceptibility was associated with the development of the ethylene signaling pathway, which suppressed the activation of the primary response CK gene of the RR factor. This led to inhibition of the expression of CK-dependent genes. Our results clearly indicate antagonistic interaction between SA and ethylene in the wheat–*S. nodorum* pathosystem, similar to what has been shown for *Arabidopsis thaliana* infected with *Pseudomonas syringae* pv tomato DC3000 [16]. We discovered that the ethylene–SA antagonism was implemented through the ethylene impact on the CK. At the same time, this does not exclude ethylene effect on the SA, because the negative action of ethylene signaling on SA biosynthesis has been shown [16]. Our results also suggest cytokinin–SA synergism, which follows from a similar plant response to SA or *trans*-zeatin treatment during infection with *S. nodorum* (Figure 5, Figure 6 and Figure 7). The synergistic interaction between SA and CK has been shown for some pathosystems [17,88,97,102]. These studies have presented that an increase in CK levels and CK signaling activation leads to an enhanced immunity by the acceleration of the defense plant response through the SA pathway. The type-B cytokinin response regulator ARR2 is known to interact with SA signalling pathway protein TGA3 (a bZIP transcription factor), which induces expression of defense related genes by binding to their promoters [17,113,114]. Thus, cytokinin-activated type-B RRs play a significant role in hormone crosstalk [112].

We found that NE SnTox3 induced biosynthesis and the signaling pathway of ethylene, which suppressed CK and SA signaling at the early stages of infection. This is indicated by the inhibition of the transcription of *TaWRKY13* and *TaRR21* genes in the SnTox3-sensitive Kaz10 plants and plants treated with ET and infected with virulent SnB (Figure 9). In contrast, treatment with either SA or *trans*-zeatin induced *TaWRKY13* and *TaRR21* gene transcription and suppressed ethylene signaling. This is indicated by the inhibition of EIN3 (*TaEIL1*) and ERF1 (*TaPIE1*) gene transcription in the SnTox3-insensitive Om35 plants and plants treated with either SA or *trans*-zeatin and infected with virulent SnB (Figure 9). Our results are consistent with some studies, which showed that CK treatment regulated the expression of the TFs WRKY and ERF [115,116]. Some WRKYs can control the expression of NPR1, required for CK- and SA-mediated defense crosstalk, as npr1 mutants did not show increased CK resistance upon exogenous application of *trans*-zeatin [89]. We found that the increase in transcription levels of the *TaWRKY13* and *TaRR21* genes preceded the increase in mRNA level of the *PR-1* and *PR-2* genes in the SnTox3-insensitive Om35 plants and plants treated with SA or *trans*-zeatin and infected with virulent SnB (Figure 7 and Figure 9), which coincides with the data of other authors [117]. Thereby, our results suggest that *TaWRKY13*, *TaRR21*, EIN3 (*TaEIL1*), and ERF1 (*TaPIE1*) are hubs of the ethylene–SA–CK crosstalk in the wheat–*S. nodorum* pathosystem. In summary, NE SnTox3 activated ethylene signaling and suppressed CK signaling, which increased the susceptibility of wheat to *S. nodorum*. Thus, crosstalk between ethylene and cytokinins determines resistance/susceptibility of wheat plants to *S. nodorum*.

## 5. Conclusions

We showed that SnTox3 inhibited PTI responses, including ROS production and SA-dependent defense responses by manipulating ethylene signaling in plants infected with *S. nodorum*. In turn, the ethylene signaling pathway affected the enzymes of redox metabolism by suppressing the oxidative burst and thereby creating favorable conditions for the development of the pathogen. Furthermore, the ethylene signaling pathway interfered with the cytokinin signaling pathway, while the latter induced an oxidative burst and SA-dependent defense response and ultimately led to the development of PTI. For the first time, we have shown that NE SnTox3 inhibits biosynthesis, modulates metabolism, and activates oxidative degradation of cytokinins in ethylene-dependent and ethylene-independent manners at the early stages of infection, which shows the critical role of CK in the regulation of plant immunity. It is worth paying attention to the fact that in the case of inhibition of biosynthesis and activation of degradation of cytokinin, SnTox3 could act independently of ethylene. Further experiments will help identify these mechanisms, which may be associated with crosstalk of the cytokinins with other hormonal signaling pathways. Further study of the mechanisms of hormone crosstalk in the wheat–*S. nodorum* pathosystem will help to reveal the general mechanisms of resistance/susceptibility of plants to NE producing pathogens. Such information will make an important contribution to the development of Marker-Associated Breeding (MAB).

## Figures and Tables

**Figure 1 biomolecules-11-00174-f001:**
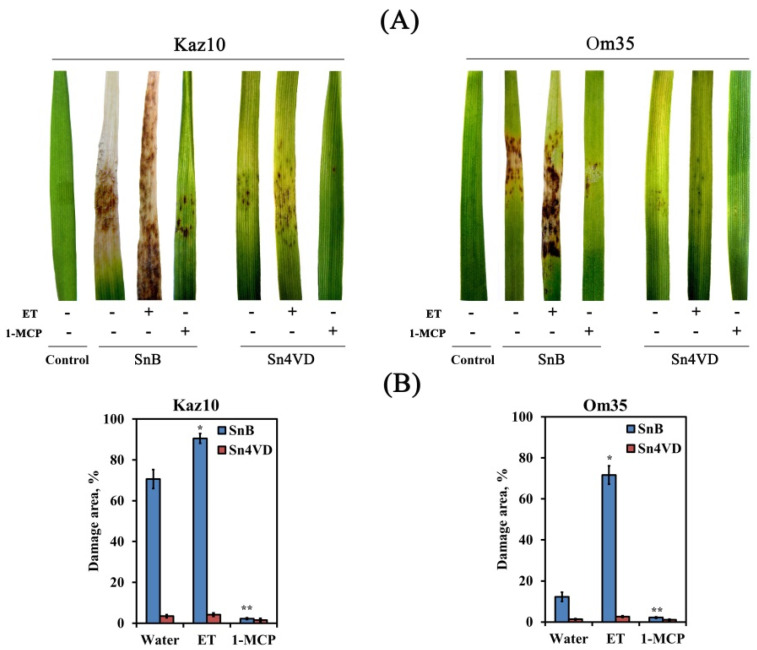
Reaction of Kazakhstanskaya 10 (Kaz10) and Omskaya 35 (Om35) wheat cultivars when inoculaed with virulent SnB and avirulent Sn4VD *S. nodorum* isolates after treatment with either ethephone (ET) or 1-methylcyclopropene (1-MCP). (**A**) Septoria nodorum blotch symptoms 8 days after inoculation. Photographs represent results of a typical variant from a series of experiments. (**B**) Damage zones on the leaves of Kaz10 and Om35 on the eighth day of inoculation, expressed as percent of the total leaf area. The samples are indicated as follows: Control—plants that have neither been treated nor inoculated with *S. nodorum* isolates; water—plants that have been sprayed with water; ET—plants that have been treated with ethephone (ET), releasing ethylene; 1-MCP—plants that have been treated with 1-methylcyclopropene (1-MCP), which is able to bind to ethylene receptors to block ethylene perception of. Figures present means ± SE (n = 30). Asterisks indicate means statistically different from the plants infected with *S. nodorum* without any pretreatments and different number of asterisks (* and **) allows distinguishing of significantly different variants according to the LSD test at * *p* < 0.05, ** *p* < 0.01.

**Figure 2 biomolecules-11-00174-f002:**
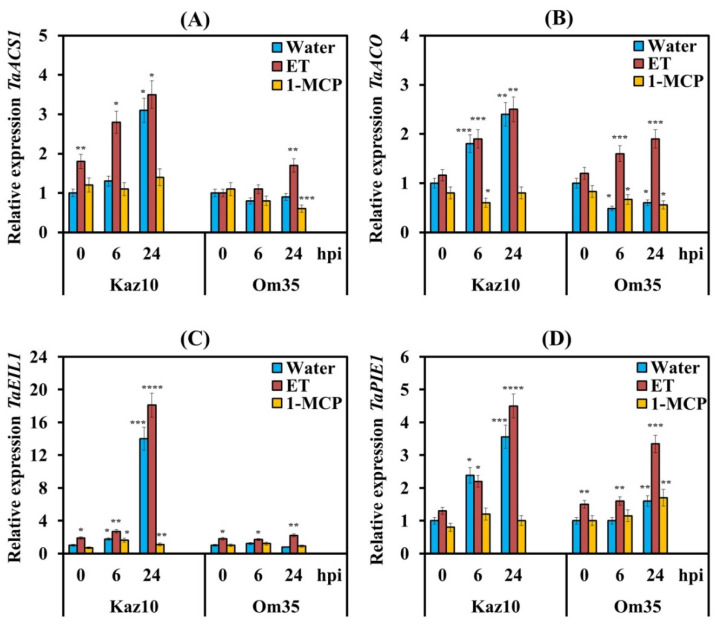
Pretreatment with ethephone (ET) upregulated and pretreatment with 1-MCP downregulated the relative expression of genes involved in the biosynthesis and signaling pathway of ethylene at the early stages of infection of Kaz10 and Om35 wheat cultivars infected with SnB. (**A**) The mRNA level of the *TaACS1* gene. (**B**) *TaACO* gene. (**C**) *TaEIL1* gene. (**D**) *TaPIE1* gene. Expression values were normalized to the housekeeping gene *TaRLI* as an internal reference and expressed relative to the normalized expression levels in mock-treated control (water) plants at 0 hpi. The samples are indicated as follows: water—noninfected plants (0 hpi) or plants infected with the *S. nodorum*, sprayed in water; ET—noninfected plants (0 hpi) or plants infected with *S. nodorum*, treated with ethephone (ET); 1-MCP—uninfected plants (0 hpi) or plants infected with *S. nodorum,* treated with 1-methylcyclopropene (1-MCP). Figures present means ± SE (n = 6). Asterisks indicate means statistically different from the control and different number of asterisks (*, **, ***, ****) allows distinguishing of significantly different variants according to the LSD test at * *p* < 0.05, ** *p* < 0.01, *** *p* < 0.001, **** *p* < 0.0001.

**Figure 3 biomolecules-11-00174-f003:**
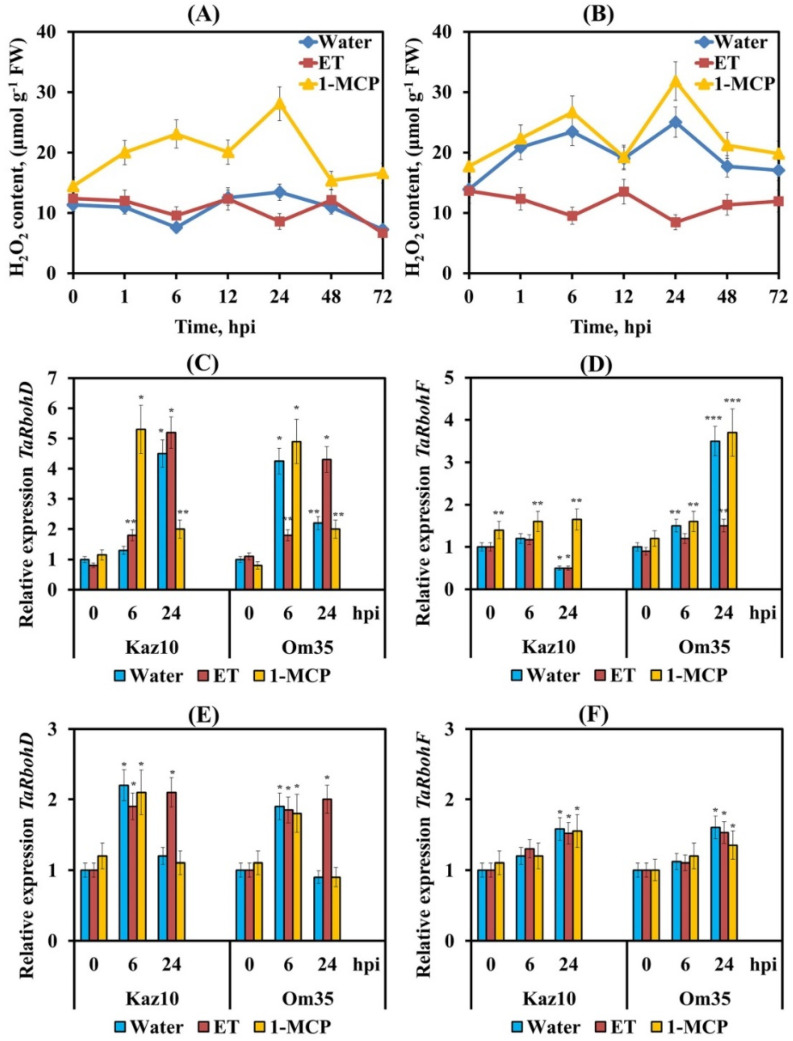
The effect of pretreatment with ethephone (ET) and 1-MCP on the H_2_O_2_ production and relative expression of genes encoding isozymes of NADPH oxidase in Kaz10 and Om35 wheat cultivars infected with virulent SnB and avirulent Sn4VD *S. nodorum* isolates. (**A**) H_2_O_2_ production in Kaz10 infected with SnB in the dynamics of the infectious process. (**B**) H_2_O_2_ production in the Om35 infected with SnB in the dynamics of the infectious process. (**C**) The mRNA abundance of the *TaRbohD* in Kaz10 and Om35 infected with SnB. (**D**) The mRNA abundance of the *TaRbohF* gene in Kaz10 and Om35 infected with SnB. (**E**) The mRNA abundance of the *TaRbohD* gene in the Kaz10 and Om35 infected with Sn4VD. (**F**) The mRNA abundance of the *TaRbohF* gene in the Kaz10 and Om35 infected with Sn4VD. Expression values were normalized to the housekeeping gene *TaRLI* as an internal reference and expressed relative to the normalized expression levels in mock-treated control (Water) plants at 0 hpi. Symbols are the same as in Figure 2. Figures present means ± SE (n = 6). Asterisks indicate means statistically different from the control and different number of asterisks (*, **, ***) allows distinguishing of significantly different variants according to the LSD test at * *p* < 0.05, ** *p* < 0.01, *** *p* < 0.001.

**Figure 4 biomolecules-11-00174-f004:**
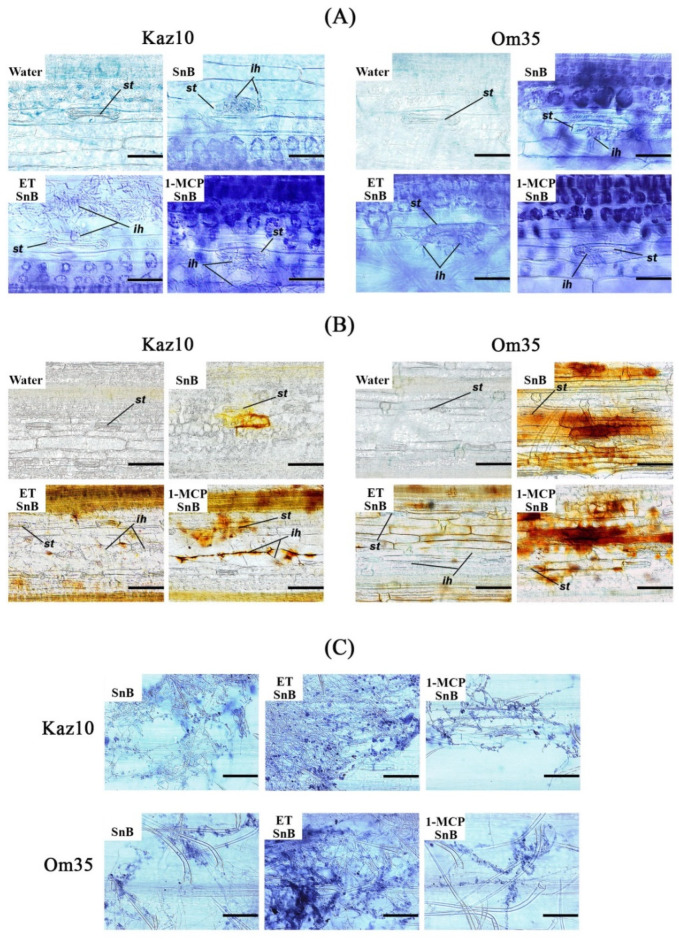
The effect of pretreatment with ethephone (ET) and 1-MCP on local generation of superoxide radical and H_2_O_2_ and growth of pathogen mycelium in leaves of Kaz10 and Om35 wheat cultivars infected with SnB. (**A**) Superoxide radical accumulation in infected leaves of Kaz10 and Om35 at 6 hpi detected with HBT staining. Bar = 50 µm. (**B**) H_2_O_2_ accumulation in infected leaves of Kaz10 and Om35 at 24 hpi detected with diaminobenzidine (DAB) staining. Bar = 100 µm. (**C**) The development of the *S. nodorum* mycelium on the leaves of Kaz10 and Om35 at 24 hpi detected with aniline blue staining. Bar = 50 µm. The variants are numbered as follows: water—noninfected plants; SnB—plants infected with the *S. nodorum* SnB; ETSnB—plants infected with *S. nodorum* SnB, treated with ethephone (ET); 1-MCPSnB—plants infected with *S. nodorum* SnB, treated with 1-methylcyclopropene (1-MCP). Photographs represent results of a typical variant from a series of experiments (n = 10). ***ih***—infections hyphae; ***st***—stomata.

**Figure 5 biomolecules-11-00174-f005:**
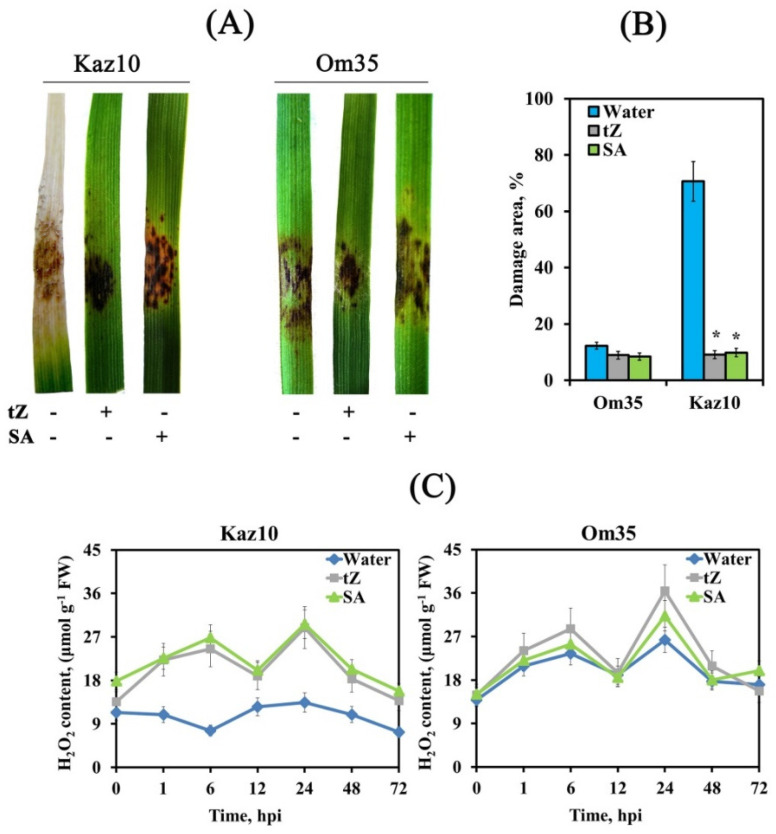
The pretreatment of *trans*-zeatin (tZ) and Salicylic acid (SA) induced an oxidative burst and reduced the damage zones on the leaves Kaz10 and Om35 cultivars infected with SnB. (**A**) Septoria nodorum blotch symptoms 8 days after inoculation. Photographs represent results of a typical variant from a series of experiments. (**B**) Damage zones on the leaves of Kaz10 and Om35 on the eighth day of inoculation, expressed as percentage of the total leaf area. Figures present means ± SE (n = 30). (**C**) H_2_O_2_ production in Kaz10 and Om35 infected with SnB in the dynamics of the infectious process. Figures present means ± SE (n = 6). The variants are indicated as follows: water—noninfected plants (0 hpi) or plants infected with the *S. nodorum*, which have been sprayed with water; tZ—noninfected plants (0 hpi) or plants infected with *S. nodorum,* treated with *trans*-zeatin; SA—uninfected plants (0 hpi) or plants infected with *S. nodorum,* treated with salicylic acid. Asterisks indicate means statistically different from the control group infected with *S. nodorum* without any pretreatments according to the LSD test at *p* < 0.05.

**Figure 6 biomolecules-11-00174-f006:**
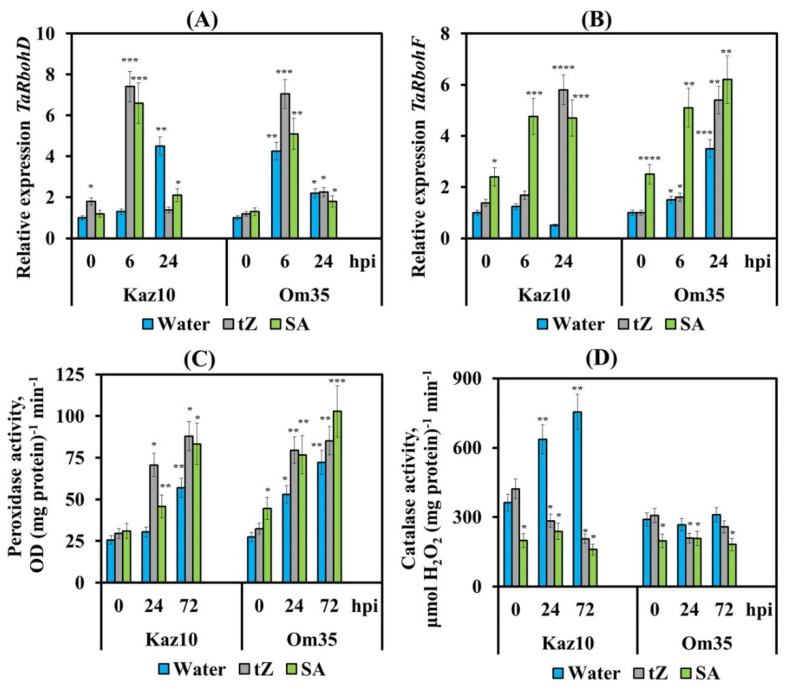
The effect of pretreatment with tZ and SA on the relative expression of genes encoding NADPH oxidase, and activity of peroxidase and catalase in two wheat cultivars of in Kaz10 and Om35 infected with SnB. (**A**) The mRNA abundance of the *TaRbohD* gene in the Kaz10 and Om35 at 6 and 24 hpi. (**B**) The mRNA abundance of the *TaRbohF* gene in the two Kaz10 and Om35 cultivars at 6 and 24 hpi. Expression values were normalized to the housekeeping gene *TaRLI* as an internal reference and expressed relative to the normalized expression levels in mock-treated control (water) plants at 0 hpi. (**C**) Peroxidase activity in the Kaz10 and Om35 cultivars at 24 and 72 hpi. (**D**) Catalase activity in the Kaz10 and Om35 cultivars at 24 and 72 hpi. Symbols are the same as in Figure 5. Figures present means ± SE (n = 6). Asterisks indicate means statistically different from the control and different number of asterisks (*, **, ***, ****) allows distinguishing of significantly different variants according to the LSD test at * *p* < 0.05, ** *p* < 0.01, *** *p* < 0.001, **** *p* < 0.0001.

**Figure 7 biomolecules-11-00174-f007:**
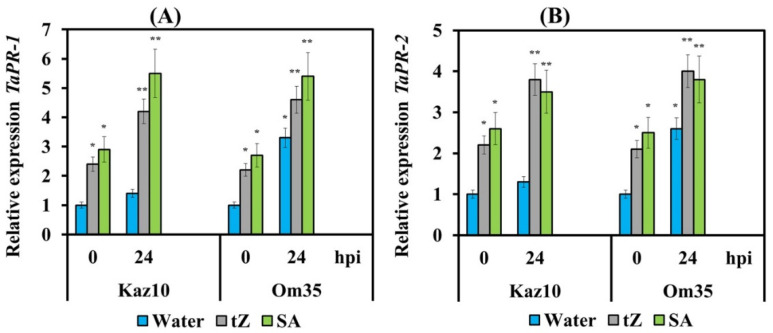
The effect of pretreatment with tZ and SA on the relative expression of salicylate signaling pathway marker genes *TaPR-1* and *TAPR-2* in Om35 and Kaz10 infected with SnB. (**A**) The mRNA abundance of the *TaPR-1* gene. (**B**) *TaPR-2* gene. Expression values are normalized to the housekeeping gene *TaRLI* as an internal reference and expressed relative to the normalized expression levels in mock-treated control (water) plants at 0 hpi. Symbols are the same as in Figure 5. Figures present means ± SE (n = 6). Asterisks indicate means statistically different from the control and different number of asterisks (*, **) allows distinguishing of significantly different variants according to the LSD test at * *p* < 0.05, ** *p* < 0.01.

**Figure 8 biomolecules-11-00174-f008:**
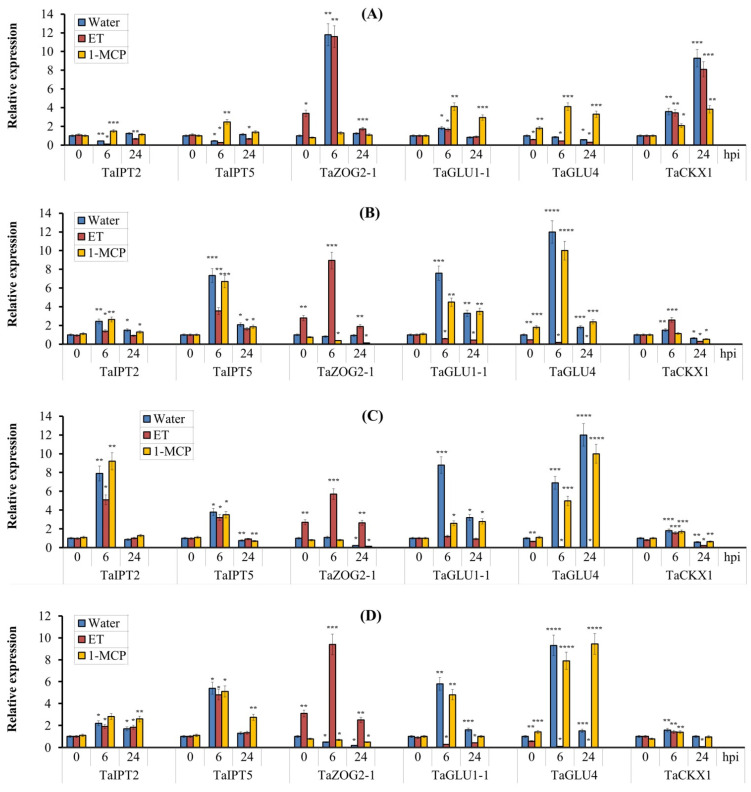
The effect of pretreatment with ethephone (ET) and 1-MCP on the relative expression of genes involved in the biosynthesis (*TaIPT2*, *TaIPT5*), O-glucosylation (*TaZOG2-1*), deglucosylation (*TaGLU1-1*, *TaGLU4*), and oxidative degradation (*TaCKX1*) of cytokinins in two wheat cultivars of Kaz10 (**A**,**C**) and Om35 (**B**,**D**) infected with virulent SnB (**A**,**B**) and avirulent Sn4VD *S. nodorum* isolates (**C**,**D**). Expression values were normalized to the housekeeping gene *TaRLI* as an internal reference and expressed relative to the normalized expression levels in mock-treated control (water) plants at 0 hpi. Symbols are the same as in Figure 2. Figures present means ± SE (n = 6). Asterisks indicate means statistically different from the control and different number of asterisks allows (*, **, ***, ****) distinguishing of significantly different variants according to the LSD test at * *p* < 0.05, ** *p* < 0.01, *** *p* < 0.001, **** *p* < 0.0001.

**Figure 9 biomolecules-11-00174-f009:**
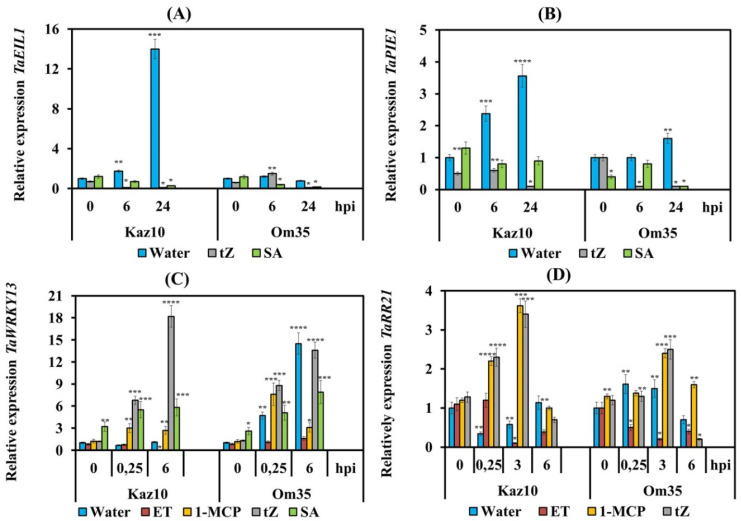
Interaction of signaling pathways of ethylene, cytokinin (CK) and SA in two wheat cultivars of Kaz10 and Om35 infected with virulent SnB. (**A**) The effect of pretreatment with tZ and SA on the relative expression of the *TaEIL1* gene. (**B**) *TaPIE1* gene. (**C**) The effect of pretreatment with ET, 1-MCP, tZ, and SA on the relative expression of the *TaWRKY13* gene. (**D**) *TaRR21* gene. Expression values are normalized to the housekeeping gene *TaRLI* as an internal reference and expressed relative to the normalized expression levels in mock-treated control (water) plants at 0 hpi. Symbols are the same as in Figure 2 and Figure 5. Figures present means ± SE (n = 6). Asterisks indicate means statistically different from the control and different number of asterisks allows (*, **, ***, ****) distinguishing of significantly different variants according to the LSD test at * *p* < 0.05, ** *p* < 0.01, *** *p* < 0.001, **** *p* < 0.0001.

**Table 1 biomolecules-11-00174-t001:** The effects of pretreatment with ethephone (ET) and 1-MCP on the content of cytokinins (pmol/g FW) in leaves of two Omskaya 35 and Kazakhstanskaya 10 cultivars at 24 h after inoculation with virulent *S. nodorum* isolate SnB.

Variety	Treatments	The Most Biologically Active Forms and Inactive Forms of CKs						
		Z	ZR	iP	iPR	Z-9G	Z-OG	Amount of CK Active Forms (Z, ZR, iP, iPR)
**Kaz10**	Water	40.5 ± 3.2	11.7 ± 0.9	5.1 ± 0.4	24.9 ± 1.9	40.9 ± 3.2	11.4 ± 0.9	82.2 ± 6.5
	SnB	30.3 ± 2.4 *	11.2 ± 0.9	14.1 ± 1.1 *	12.4 ± 0.9 *	63.1 ± 5.0 *	23.3 ± 1.8 *	67.8 ± 5.4 *
	ET	36.5 ± 2.9	15.7 ± 1.3	8.3 ± 0.7	18.5 ± 1.4	63.1 ± 5.0 *	39.9 ± 3.1 **	78.8 ± 6.3 *
	ET + SnB	32.3 ± 2.6 *	16.1 ± 1.3	17.1 ± 1.4 *	14.2 ± 1.1 *	68.4 ± 5.4 *	36.2 ± 2.9 **	79.7 ± 6.3 *
	1-MCP	38.9 ± 3.1	14.1 ± 1.1	7.1 ± 0.5	21.3 ± 1.7	39.7 ± 3.1 **	8.9 ± 0.7	81.5 ± 6.5
	1-MCP + SnB	60.6 ± 4.8 **	32.9 ± 2.6 *	33.6 ± 2.6 **	36.9 ± 2.9 **	34.1 ± 2.7 **	6.6 ± 0.5 ***	164.0 ± 13.1 **
**Om35**	Water	28.9 ± 2.3	13.7 ± 1.1	8.2 ± 0.6	22.4 ± 1.7	54.7 ± 4.3	3.7 ± 0.2	73.3 ± 5.8
	SnB	82.6 ± 6.1 *	38.6 ± 3.0 *	37.7 ± 3.0 *	49.8 ± 3.9 *	43.8 ± 3.5 *	0.8 ± 0.6 *	208.8 ± 16.7 *
	ET	36.4 ± 2.9 **	13.1 ± 1.0	9.2 ± 0.7	23.0 ± 1.8	55.5 ± 4.4	13.5 ± 1.0 **	81.8 ± 6.5 **
	ET + SnB	30.4 ± 2.4	17.1 ± 1.3	33.6 ± 2.6 *	34.8 ± 2.7 **	66.8 ± 5.3 **	45.1 ± 3.6 ***	116.0 ± 9.2 ***
	1-MCP	33.1 ± 2.6	13.4 ± 1.0	10.9 ± 0.8	28.4 ± 2.2	39.2 ± 3.1 *	9.9 ± 0.7 **	85.9 ± 6.8 **
	1-MCP + SnB	85.8 ± 6.9 *	40.4 ± 3.2 *	31.9 ± 2.5 *	56.8 ± 4.5 *	26.2 ± 2.0 ***	0.9 ± 0.1 *	215.0 ± 17.2 *

Asterisks indicate means statistically different from the control in each cultivar and different number of asterisks (*, **, ***) allows distinguishing of significantly different variants according to the LSD test (n = 6, * *p* < 0.05, ** *p* < 0.01, *** *p* < 0.001).

**Table 2 biomolecules-11-00174-t002:** The effect of pretreatment with ethephone (ET) and 1-MCP on the content of cytokinins (pmol/g FW) in leaves of Omskaya 35 and Kazakhstanskaya cultivars 10 at 24 h after inoculation with *S. nodorum* avirulent isolate Sn4VD.

Variety	Treatments	The Most Biologically Active Forms and Inactive Forms of CKs						
		Z	ZR	iP	iPR	Z-9G	Z-OG	Amount of CK Active Forms (Z, ZR, iP, iPR)
**Kaz10**	Water	39.8 ± 3.1	11.6 ± 0.9	4.9 ± 0.4	22.5 ± 1.8	34.8 ± 2.7	8.8 ± 0.7	79.0 ± 6.3
	Sn4VD	70.3 ± 5.6 *	19.3 ± 1.5 *	27.3 ± 2.1 *	47.7 ± 3.8 *	42.3 ± 3.3 *	0.6 ± 0.1 *	164.8 ± 13.1 *
	ET	49.5 ± 3.9	15.2 ± 1.2	8.6 ± 0.6 **	16.7 ± 1.	43.3 ± 3.4 *	16.7 ± 1.3 **	90.2 ± 7.2
	ET + Sn4VD	63.5 ± 5.0 *	18.1 ± 1.4 *	25.2 ± 2.0 *	34.6 ± 2.7 **	47.3 ± 3.7 *	14.0 ± 1.1 **	141.5 ± 11.3 *
	1-MCP	72.7 ± 5.8 *	11.7 ± 0.9	5.1 ± 0.4	26.0 ± 2.0	43.7 ± 3.5 *	1.0 ± 0.1 *	115.6 ± 9.2 **
	1-MCP + Sn4VD	90.0 ± 7.2 **	17.6 ± 1.4 *	32.3 ± 2.5 *	52.2 ± 4.1 *	29.1 ± 2.3 **	1.4 ± 0.1 *	192.2 ± 15.3 ***
**Om35**	Water	39.7 ± 3.1	11.9 ± 0.9	9.7 ± 0.7	20.9 ± 1.6	40.9 ± 3.2	7.6 ± 0.6	82.4 ± 6.6
	Sn4VD	70.2 ± 5.6 *	24.5 ± 1.9 *	26.4 ± 2.1 *	41.5 ± 3.3 *	37.2 ± 2.9	1.0 ± 0.1*	162.8 ± 13.0 *
	ET	39.9 ± 3.2	14.8 ± 1.1	11.8 ± 0.9	20.6 ± 1.6	49.9 ± 4.0	17.7 ± 1.4 **	87.3 ± 6.9
	ET + Sn4VD	46.8 ± 3.7 **	23.5 ± 1.8 *	30.6 ± 2.4 *	42.8 ± 3.4 *	41.5 ± 3.3	9.5 ± 0.7	143.7 ± 11.5 *
	1-MCP	35.0 ± 2.8	16.5 ± 1.3	8.0 ± 0.6	17.3 ± 1.3	40.7 ± 3.2	9.9 ± 0.8	76.9 ± 6.1
	1-MCP + Sn4VD	88.3 ± 7.0 ***	28.0 ± 3.7 *	46.8 ± 3.7 **	43.7 ± 3.5 *	37.7 ± 3.0	1.4 ± 0.1 *	207.0 ± 16.5 **

Asterisks indicate means statistically different from the control in each cultivar and different number of asterisks allows (*, **, ***) distinguishing of significantly different variants according to the LSD test (n = 6, * *p* < 0.05, ** *p* < 0.01, *** *p* < 0.001).

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
