# Peer review of "Ethylene-Cytokinin Interaction Determines Early Defense Response of Wheat against Stagonospora nodorum Berk."

_biomolecules, 2021, doi:10.3390/biom11020174_

Round 1

Reviewer 1 Report

The presented manuscript is the continuation of the previous work of the Authors, which concerning roles of ethylene and cytokinins in defense response in wheat infected with Septoria nodorum (paper form 2016). In fact, the presented results are rather widening of the previous experiments with the confirmation of results obtained previously. However, I strongly encourage to reduce the text volume of the manuscript. The section Results should contain results only, without speculations, suggestions and some kind of introduction (e.g. Lines 634 – 646, and 668 – 671, and 683 – 694, and 706 – 720, and 740 – 745). The presented Discussion is too verbose (e.g. Lines 926 – 962, and 966 – 992, 1084 – 1106). But after modification can be the good basis for review article.

Reviewer 2 Report

The manuscript by Veselova et al. describes an analysis of ethylene-cytokinin interaction in wheat responses to the pathogen Stagonospora nodorum by using cultivars differing in resistance and fungal isolates with or without virulence (dependent on presence of the effector SnTox3). The work encompasses a variety of relevant, well-established methodologies and experiments, ranging from chemical treatments to qPCR analysis, that quite effectively allow them to conclude on their findings, albeit without much innovation. I have very few comments which are all regarding wording - especially in the introduction, but am sure that these will be picked up upon further proofreading and therefore find it unnecessary to point out specifics. Overall, I wish to congratulate the authors on a nice job and recommend the paper for publication in Biomolecules as is.

Author Response

The authors are grateful to the Reviewer for positive comments on our work.

Reviewer 3 Report

The article “Ethylene-Cytokinin Interaction Determines Early Defense Response of wheat against Stagonospora nodorum Berk” contains original and very interesting results. However, the article needs to be improved in the presentation of the text and figures, and some reduction of the text.

My comments are as follows:

Lines 24-26: Several mechanisms are listed, but it is not clear which one is of prime importance.

Line 43: Is this abbreviation R correct?

Line 53: Please, use abbreviation NEs.

Lines 99-101: Should this text be partially in lower case letters?

Line 102: Abbreviation WRKY should be decrypted.

Subsection 2.1: Please, add the source of wheat seeds and fungal strains.

Lines 279-283, 296-298: This is repetition of information from Materials and Methods section. I suggest removing this text.

Figure 1: Typical leaf of the control (uninoculated and untreated) variant for both cultivars should be added to parts A and B. 

Lines 328-334: This sentence is confusing and needs to be improved and divided into several sentences.

Figure 4: This drawing is complex. You have to look too often at the caption to the picture. Therefore, I recommend that instead of the numbers (1, 2, 3 and 4) indicate directly in the figure the designations of the treatments for the experiment. For example, instead of 1, specify uninfected; instead of 2, specify SnB; … and so on…

Tables 1 and 2 need to be formatted so that each treatment fits on single line.

Figure 8: Duplicate data as numbers is unnecessary. Please, remove it. It is not clear where wheat varieties Kaz10 and Om35 are located on this figure.

Lines 361-365, 371-373, 396-405, 428-432, 626-629, 635-637, 642-647, 665-671, 684-689, 691-694, 707-720 740-745: This text belongs to the Discussion section. I suggest removing it.

In general, the Results section is too long and the text should be shortened by describing only the most important results. The Discussion section is well written, but also very long and too much detailed. I also suggest this text should be shortened.

Lines 776-777: Please, give reference(s) to this statement.

Line 870: Please, clarify zeatin or trans-zeatin?

Line 1118: “in some cases…” - Can you clarify in which cases?

Round 2

Reviewer 3 Report

The authors carefully answered all questions and made the necessary changes in the article.

The only minor comment is to Line 138: If you use the word “obtained” between words “virulent” and “from”, you do not need using repeating brackets.